



# ISMIP6 Antarctica: a multi-model ensemble of the Antarctic ice sheet evolution over the 21st century

Hélène Seroussi [1], Sophie Nowicki [2], Antony J. Payne [3], Heiko Goelzer [4,5], William H. Lipscomb [6], Ayako Abe Ouchi [7], Cécile Agosta [8], Torsten Albrecht [9], Xylar Asay-Davis [10], Alice Barthel [10], Reinhard Calov [9], Richard Cullather [2], Christophe Dumas [8], Rupert Gladstone [11], Nicholas Golledge [12], Jonathan M. Gregory [13,14], Ralf Greve [15], Tore Hatterman [16,17], Matthew J. Hoffman [10], Angelika Humbert [18,19], Philippe Huybrechts [20], Nicolas C. Jourdain [21], Thomas Kleiner [18], Eric Larour [1], Gunter R. Leguy [6], Daniel P. Lowry [22], Chistopher M. Little [23], Mathieu Morlighem [24], Frank Pattyn [5], Tyler Pelle [24], Stephen F. Price [10], Aurélien Quiquet [8], Ronja Reese [9], Nicole-Jeanne Schlegel [1], Andrew Shepherd [25], Erika Simon [2], Robin S. Smith [13], Fiammetta Straneo [26], Sainan Sun [5], Luke D. Trusel [27], Jonas Van Breedam [19], Roderik S. W. van de Wal [4,28], Ricarda Winkelmann [9,29], Chen Zhao [30], Tong Zhang [10], and Thomas Zwinger [31]

[1]Jet Propulsion Laboratory, California Institute of Technology, Pasadena, CA, USA
[2]NASA Goddard Space Flight Center,Greenbelt, MD, USA
[3]University of Bristol, United Kingdom
[4]Institute for Marine and Atmospheric research Utrecht, Utrecht University, The Netherlands
[5]Laboratoire de Glaciologie, Université Libre de Bruxelles, Brussels, Belgium
[6]Climate and Global Dynamics Laboratory, National Center for Atmospheric Research, Boulder, CO, USA
[7]University of Tokyo, Japan
[8]Laboratoire des sciences du climat et de l'environnement, LSCE-IPSL, CEA-CNRS-UVSQ, Université Paris-Saclay, France
[9]Potsdam Institute for Climate Impact Research (PIK), Member of the Leibniz Association, P.O. Box 60 12 03, 14412 Potsdam, Germany
[10]Theoretical Division, Los Alamos National Laboratory, NM, USA
[11]Arctic Centre, University of Lapland, Finland
[12]Antarctic Research Centre, Victoria University of Wellington, New Zealand
[13]National Centre for Atmospheric Science, University of Reading, United Kingdom
[14]Met Office Hadley Centre, Exeter, United Kingdom
[15]Institute of Low Temperature Science, Hokkaido University, Sapporo, Japan
[16]Norwegian Polar Institute, Tromsø, Norway
[17]Energy and Climate Group, Department of Physics and Technology, The Arctic University – University of Tromsø, Norway
[18]Alfred Wegener Institute for Polar and Marine Research, Bremerhaven, Germany
[19]Department of Geoscience, University of Bremen, Bremen, Germany
[20]Earth System Science and Departement Geografie, Vrije Universiteit Brussel, Brussels, Belgium
[21]Univ. Grenoble Alpes/CNRS/IRD/G-INP, Institut des Géosciences de l'Environnement, France
[22]GNS Science, Lower Hutt, New Zealand
[23]Atmospheric and Environmental Research, Inc., Lexington, Massachusetts, USA
[24]Department of Earth System Science, University of California Irvine, Irvine, CA, USA
[25]University of Leeds, Leeds, United Kingdom
[26]Scripps Institution of Oceanography, University of California San Diego, La Jolla, CA, USA
[27]Department of Geography, Pennsylvania State University, University Park, PA, USA
[28]Geosciences, Physical Geography, Utrecht University, Utrecht, the Netherlands
[29]University of Potsdam, Institute of Physics and Astronomy, Karl-Liebknecht-Str. 24-25, 14476 Potsdam, Germany
[30]University of Tasmania, Hobart, Australia





[31]CSC-IT Center for Science, Espoo, Finland

**Correspondence:** Helene Seroussi (helene.seroussi@jpl.nasa.gov)

**Abstract.** Ice flow models of the Antarctic ice sheet are commonly used to simulate its future evolution in response to different climate scenarios and inform on the mass loss that would contribute to future sea level rise. However, there is currently no consensus on estimated the future mass balance of the ice sheet, primarily because of differences in the representation of physical processes and the forcings employed. This study presents results from 18 simulations from 15 international groups focusing on the evolution of the Antarctic ice sheet during the period 2015-2100, forced with different scenarios from the Coupled Model Intercomparison Project Phase 5 (CMIP5) representative of the spread in climate model results. The contribution of the Antarctic ice sheet in response to increased warming during this period varies between -7.8 and 30.0 cm of Sea Level Equivalent (SLE). The evolution of the West Antarctic Ice Sheet varies widely among models, with an overall mass loss up to 21.0 cm SLE in response to changes in oceanic conditions. East Antarctica mass change varies between -6.5 and 16.5 cm SLE, with a significant increase in surface mass balance outweighing the increased ice discharge under most RCP 8.5 scenario forcings. The inclusion of ice shelf collapse, here assumed to be caused by large amounts of liquid water ponding at the surface of ice shelves, yields an additional mass loss of 8 mm compared to simulations without ice shelf collapse. The largest sources of uncertainty come from the ocean-induced melt rates, the calibration of these melt rates based on oceanic conditions taken outside of ice shelf cavities and the ice sheet dynamic response to these oceanic changes. Results under RCP 2.6 scenario based on two CMIP5 AOGCMs show an overall mass loss of 10 mm SLE compared to simulations done under present-day conditions, with limited mass gain in East Antarctica.

# 1 Introduction

Remote sensing observations of the Antarctic ice sheet have shown continuous ice mass loss over at least the past four decades (Rignot et al., 2019; Shepherd et al., 2019, 2018), in response to changes in oceanic (Thomas et al., 2004; Jenkins et al., 2010) and atmospheric (Vaughan and Doake, 1996; Scambos et al., 2000) conditions. This overall mass loss has large spatial variations, as regions around Antarctica experience varying climate change patterns, and individual glaciers may respond differently to similar forcings depending on their local geometry and internal dynamics (Morlighem et al., 2019b). To date, the Amundsen and Bellingshausen Sea sectors of West Antarctica as well as the Antarctic Peninsula have experienced significant mass loss, while East Antarctica has had a limited response to climate change so far (Paolo et al., 2015; Gardner et al., 2018; Rignot et al., 2019).

Despite the rapid increase in the number of observations (e.g. Rignot et al., 2019; Gardner et al., 2018) and a paradigm shift in numerical ice flow models over the past decade (Goelzer et al., 2017; Pattyn et al., 2017), the uncertainty in the Antarctic ice sheet contribution to sea level over the coming centuries remains high (Ritz et al., 2015; DeConto and Pollard, 2016; Edwards





et al., 2019). Understanding past changes is critical in order to improve projections of Antarctic ice sheet evolution over the next decades and centuries in response to climate change. Previous modeling studies showed variable Antarctic contribution to sea level rise over the coming century, depending on the physical processes included (e.g., Edwards et al., 2019), forcing used (e.g., Golledge et al., 2015; Schlegel et al., 2018) or model parameterizations (e.g., Bulthuis et al., 2019), leading to results varying between a few mm to more than 1 meter of sea level contribution by the end of the century (Ritz et al., 2015; Pollard et al., 2015;

Little et al., 2013; Levermann et al., 2014). Model intercomparison efforts such as Ice2Sea (Edwards et al., 2014) and SeaRISE (Sea-level Response to Ice Sheet Evolution, Bindschadler et al., 2013; Nowicki et al., 2013a) highlighted the large discrepancies in numerical ice flow model results, even when similar climate conditions are applied for model forcing. Furthermore, most of these experiments were carried out under extremely simplified climate forcings, limiting our understanding of how ice sheets may respond to realistic climate scenarios.

ISMIP6 (Ice Sheet Model Intercomparison Project for CMIP6, Nowicki et al., 2016) is the primary effort of CMIP6 (Climate Model Intercomparison Project Phase 6) focusing on ice sheets and was designed to mitigate this gap as well as improve our understanding of ice sheet–climate interactions. In a first stage, ice sheet model initialization experiments (initMIP, Goelzer et al., 2018; Seroussi et al., 2019) focused on the role of initial conditions and model parameters in ice flow simulations. Antarctic experiments were based on idealized surface mass balance (SMB) and ocean-induced basal melt forcings to assess

the response of ice flow models to anomalies in these external forcings (Seroussi et al., 2019). Results showed that models respond similarly to changes in SMB, while changes in ocean-induced basal melt cause a large spread in model response. Treatment of sub-ice-shelf basal melt, along with model spatial resolution close to the grounding line, were identified as the main sources of differences in the simulations (Seroussi et al., 2019).

In this study, we focus on projections of the Antarctic ice sheet forced by output from CMIP5 Atmosphere-Ocean General

Circulation Models (AOGCMs) under different climate conditions, as CMIP6 results were not available when the experimental protocol was designed (Nowicki et al., in review). The ensemble of simulations focuses mostly on the 2015–2100 period and is based on 21 sets of ice flow simulations submitted by 13 international institutions. We investigate the relative role of AOGCM forcings, Representative Concentration Pathway (RCP) scenarios, ocean-induced melt parameterizations, and simulated physical processes on the Antarctic ice sheet contribution to sea level and the associated uncertainties. We first describe

the experiment set-up and the forcings used for the simulations in section 2. We then detail the ice flow models that took part in this intercomparison and summarize their main characteristics in section 3. Section 4 analyzes the results and assesses the impact of the different proposed scenarios and parameterizations. Finally, we discuss the results, differences between models, and the main sources of uncertainties in section 5.

## 2   Experiments and model set-up

ISMIP6 is an endorsed MIP (Model Intercomparison Project) of CMIP6, and experiments performed as part of ISMIP6 projections are therefore based on outputs from AOGCMs taking part in CMIP. As results from CMIP6 were not available at the time the experimental protocol was determined (Nowicki et al., in review), it was decided to rely primarily on available CMIP5





outputs to assess the future evolution of the Greenland (Goelzer et al., sub.) and Antarctic ice sheets. This choice required an in-depth analysis of CMIP5 AOGCM outputs and the selection of a subset of CMIP5 models that would capture the spread of climate evolution. The choice of using only a subset of AOGCMs limits the number of simulations required from each ice sheet modeling group, while still sampling the uncertainty in future ice sheet evolution associated with variations in climate models (Barthel et al., in review). Additional simulations based on CMIP6 are ongoing and will be the subject of a forthcoming publication.

In this section, we summarize the experimental protocol for ISMIP6-Antarctica Projections, including the choice of CMIP5 models, the processing of their outputs in order to derive atmospheric and oceanic forcings applicable to ice sheet models, and the processes included in the experiments. We then list the experiments analyzed in the present manuscript. More details on the experimental protocol can be found in (Nowicki et al., in review), while the selection protocol used to build the CMIP5 model ensemble is explained in Barthel et al. (in review). A detailed description of the ocean melt parameterization and calibration is available in Jourdain et al. (under review).

## 2.1 Forcing

### 2.1.1 Choice of AOGCMs

The forcings applied to ISMIP6-Antarctica projections are derived from both RCP 8.5 and RCP 2.6 scenarios, with most experiments based on RCP 8.5, in order to estimate the full extent of changes possible by 2100 with varying AOGCMs forcings. A few RCP 2.6 scenarios are used to assess the response of the ice sheet to moderate climate changes.

After selecting AOGCM models that performed both RCP 8.5 and RCP 2.6 scenarios, the models were first assessed on their ability to represent present climate conditions around the Antarctic ice sheet. A historical bias metric was computed, incorporating atmosphere and surface oceanic conditions south of 40° South and oceanic conditions in six ocean sectors shallower than 1500 m around Antarctica. Atmospheric and surface metrics were evaluated against the European Centre for Medium-Range Weather Forecasts "Interim" re-analysis (ERA-Interim, Dee et al., 2011). Ocean metrics were compared to a reference climatology combining the 2018 World Ocean Atlas (Locarnini et al., 2019), EN4 ocean climatology (Good et al., 2013) and temperature profiles from Logger–equipped seals (Roquet et al., 2018). Following this assessment of AOGMCs, we analyzed projected changes between 1980-2000 and 2080-2100 in oceanic and atmospheric conditions under the RCP 8.5 scenario. We chose six AOGCMs which performed better than the median at capturing present-day conditions and which represented a large diversity in projected changes. These models are CCSM4, MIROC-ESM-CHEM and NorESM1-M for the core experiments, and CSIRO-Mk3-6-0, HadGEM2-ES and IPSL-CM5A-M for the CMIP5 Tier 2 experiments (see section 2.2). Two of these models, NorESM1-M and IPSL-CM5A-M, were also chosen to provide forcings for the RCP 2.6 scenario. We refer to Barthel et al. (in review) for a detailed description of the model evaluation and selection.

This choice of AOGCMs was designed both to select models that best capture the variables relevant to ice sheet evolution and to maximize the diversity in projected 21$^{st}$ climate evolution, while limiting the number of simulations. AOGCM choices were made independently for Greenland and Antarctica, to focus on the specificities of each ice sheet and region. We derived external





forcings for the Antarctic ice sheet from these AOGCMs outputs and provided yearly forcing anomalies for participating models.

### 2.1.2 Atmospheric forcing

Using the AOGCMs selected, atmospheric forcings were derived in the form of yearly averaged surface mass balance anomalies
and surface temperature anomalies compared to the 1980-2000 period. The SMB anomalies include changes in precipitation, evaporation, sublimation, and runoff, and are presented in the form of water-equivalent quantities. These anomalies are then added to reference surface mass balance (Seroussi et al., 2019) and surface temperature fields that are used as a baseline in the ice models.

SMB conditions are often estimated using Regional Climate Models (RCMs), such as the Regional Atmospheric Climate
Model (RACMO, Lenaerts et al., 2012; van Wessem et al., 2018) and Modèle Atmosphérique Régional (MAR, Agosta et al., 2019) forced at their boundaries with AOGCMs outputs. As high-resolution RCM integrations for the full Antarctic Ice Sheet are complex and typically require additional boundary forcing and considerable time and computational resources, it was decided not to follow this approach for ISMIP6-Antarctica Projections, but to use AOGCM outputs directly. Further details on the derivation of atmospheric forcing can be found in Nowicki et al. (in review).

### 2.1.3 Oceanic forcing

Similar to what is done for the atmospheric forcing, the ocean forcing is derived from the AOGCMs outputs. However, the CMIP5 models do not resolve the Antarctic continental shelf, and none includes ice shelf cavities. The first task to prepare the ocean forcing was therefore to extrapolate relevant oceanic conditions (temperature and salinity) to areas not included in AOGCM ocean models, including areas currently covered by ice that could become ice-free in the future. These areas include
sub-ice-shelf cavities and areas beneath the grounded ice sheet that could be exposed to the ocean following ice thinning and grounding line retreat. Three-dimensional fields of ocean salinity, temperature and thermal forcing were then computed as annual mean values over the 1995–2100 period. We refer to Jourdain et al. (under review) for more details on the extrapolation of oceanic fields and computation of ocean thermal forcing.

Converting ocean conditions into ocean-induced melt at the base of ice shelves is an active area of research, and several
parameterizations with different levels of complexity have recently been proposed for converting ocean conditions into ice shelf melt rates (Lazeroms et al., 2018; Reese et al., 2018a; Pelle et al., 2019). As only a limited number of direct observations of ocean conditions (Jenkins et al., 2010; Dutrieux et al., 2014) and ice shelf melt rates (Rignot et al., 2013; Depoorter et al., 2013) exist, these parameterizations are difficult to calibrate and evaluate. Some are relatively complex and based on non-local quantities, and can therefore be difficult to implement in continental-scale parallel ice sheet models. Furthermore, such
parameterizations do not account for feedbacks between the ice and ocean dynamics, which are likely only captured by coupled ice–ocean models (De Rydt and Gudmundsson, 2016; Seroussi et al., 2017; Favier et al., 2019).

For these reasons, ISMIP6-Antarctica Projections include two options that can be adopted for the sub-ice shelf melt parameterization: 1) a standard parameterization based on a prescribed relation between ocean thermal forcing and ice shelf





melting rates and 2) an open parameterization left to the discretion of the ice sheet modeling groups. Such a framework allows
us to evaluate the response to a wide spectrum of melt parameterizations with the open framework, while also capturing the
uncertainty related to the ice sheet response under a more constrained set-up in the standard framework. The standard pa-
rameterization was chosen as a trade-off between a simple parameterization that most modeling groups could implement in a
limited time, while capturing melt rate patterns as accurately as possible. Results from an idealized case comparing coupled
ice–ocean models with different melt parameterizations suggested that a non-local, quadratic melt parameterization was best
able to mimic the coupled ice–ocean results over a broad range of ocean forcing (Favier et al., 2019):

$$m\left(x,y\right) = \gamma_0 \times \left(\frac{\rho_{sw}c_{pw}}{\rho_i L_f}\right)^2 \times \left(TF\left(x,y,z_{\mathrm{draft}}\right) + \delta T_{\mathrm{sector}}\right) \times \left|\langle TF\rangle_{\mathrm{draft}\in\mathrm{sector}} + \delta T_{\mathrm{sector}}\right|, \tag{1}$$

where $\gamma_0$ is a coefficient similar to an exchange velocity, $\rho_{sw}$ the ocean density, $c_{pw}$ the specific heat of sea water, $\rho_i$ the ice
density, $L_f$ the ice latent heat of fusion, $TF(x,y,z_{\mathrm{draft}})$ the local ocean thermal forcing at the ice shelf base, $\left|\langle TF\rangle_{\mathrm{draft}\in\mathrm{sector}}\right|$
the ocean thermal forcing averaged over a sector, and $\delta T_{\mathrm{sector}}$ the temperature correction for each sector. The values for $\gamma_0$
and $\delta T_{\mathrm{sector}}$ in this equation were calibrated from observations of ocean conditions and melt rates based either on circum-
Antarctic observations (the "MeanAnt" method) or on observations close to the grounding line of Pine Island Glacier (the
"PIGL" method). The coefficient $\gamma_0$ is first calibrated assuming $\delta T$ equal to zero and using $10^5$ random samplings of Antarctic
melt rate and ocean temperature, so that the total melt produced under the ice shelves is similar to melt rates estimated in Rignot
et al. (2013) and Depoorter et al. (2013). This process provides a distribution of possible $\gamma_0$ values. The $\delta T_{\mathrm{sector}}$ values are then
calibrated for each of 16 sectors of Antarctica (see Jourdain et al., under review, for details) so that the melt in each basin agrees
with average estimated melt in this sector. The median value of $\gamma_0$ is used for all but two runs. These two experiments assess the
impact of uncertainty in $\gamma_0$ by using the 5[th]- and 95[th]-percentile values from the distribution. The second calibration, "PIGL",
uses the same process, but constrained with only a subset of observations under Pine Island ice shelf and close to its grounding
line, since these values are the most relevant for highly dynamic ice streams that have the highest sub-shelf melt (Reese et al.,
2018b). This calibration leads to higher values of $\gamma_0$, corresponding to a greater sensitivity of melt rates to changes in ocean
temperature.

The choice of melt parameterization and its calibration with observations is described in detail in Jourdain et al. (under
review). For models that could not implement such a non-local parameterization, a local quadratic parameterization similar to
Eq.1, with the non-local thermal forcing replaced by local thermal forcing, was also designed and calibrated to provide similar
results (Jourdain et al., under review).

### 2.1.4 Ice shelf collapse

Several ice shelves in the Antarctic Peninsula have collapsed over the past three decades (Doake and Vaughan, 1991; Scambos
et al., 2004, 2009). The main mechanism proposed to explain the collapse of these ice shelves is the presence of significant
amounts of liquid water on their surface, which cause hydrofracturing and ultimately lead to their collapse (Vaughan and Doake,
1996; Banwell et al., 2013; Robel et al., 2019). Shelf collapse leads to acceleration and increased mass loss of the glaciers
feeding them (De Angelis and Skvarca, 2003; Rignot et al., 2004), but more dramatic consequences have been envisioned if

ice shelves were to collapse in front of thick glaciers resting on retrograde bed slopes (Bassis and Walker, 2011; DeConto and Pollard, 2016). As the presence of liquid water at the surface of Antarctic ice shelves is expected to increase in a warming climate (Mercer, 1978; Trusel et al., 2015), we propose experiments that include ice shelf collapse. The response of grounded ice streams to such collapse is left to the discretion of individual modeling groups, and other experiments should not include ice shelf collapse.

Ice shelf collapse is described as a yearly mask that defines the regions and times of collapse. The criteria for ice shelf collapse are based on the presence of mean annual surface melting above 725 mm over a decade, similar to numbers proposed in Trusel et al. (2015), and corresponding to the average melt simulated by RACMO2 over Larsen A and B in the decade before their collapse. The amount of surface melting was computed from AOGCM surface air temperature using the methodology described in Trusel et al. (2015).

## 2.2 Experiments

The list of experiments for ISMIP6-Antarctica Projections is described and detailed in Nowicki et al. (in review). It includes a historical experiment (*historical*), control runs (*ctrl* and *ctrl_proj*), simple anomaly experiments similar to initMIP-Antarctica (*asmb* and *abmb*), 13 core (Tier 1) experiments and 8 Tier 2 experiments based on CMIP5 forcing. The list is repeated in Table 1 for completeness. In summary, these experiments include:

- 12 experiments based on RCP 8.5 scenarios from 6 AOGCMs (open and standard melt parameterizations)

- 4 experiments based on RCP 2.6 scenarios from 2 AOGCMs (open and standard melt parameterizations)

- 2 experiments including ice shelf collapse (open and standard melt parameterizations)

- 2 experiments testing the uncertainty in the melt parameterization (standard melt parameterization only)

- 2 experiment testing the uncertainty in the melt calibration (standard melt parameterizations only)

All experiments start in 2015, except for the historical, ctrl, asmb, and abmb experiments, which start at the model initialization time. The historical experiment runs from the initialization time until the beginning of 2015, while the ctrl, asmb, and abmb experiments run for either 100 years or until 2100, whichever is longer. The other experiments run to the end of 2100. The ctrl_proj run is a control run similar to ctrl: a simulation under constant climate conditions representative of the recent past. The only difference is that ctrl_proj starts in 2015.

Most analyses presented in this study follow an "experiment minus ctrl_proj" approach, so these results provide the impact of change in climatic conditions relative to ice sheets forced with present-day conditions until 2100. We know that ice sheets respond non-linearly to changes in climate conditions, but such an approach is necessary as ice flow models often do not accurately capture the trends observed over the recent past (Seroussi et al., 2019).





**Table 1.** List of ISMIP6-Antarctic Projections Core (Tier 1) experiments and Tier 2 experiments based on CMIP5 AOGCMs.

| Experiment | AOGCM | Scenario | Ocean Forcing | Ocean coefficient | Ice Shelf Fracture | Tier |
|---|---|---|---|---|---|---|
| historical | None | None | Free | Medium | No | Tier 1 (Core) |
| ctrl | None | None | Free | Medium | No | Tier 1 (Core) |
| ctrl_proj | None | None | Free | Medium | No | Tier 1 (Core) |
| asmb | None | None | Same as ctrl +SMB anomaly | Medium | No | Tier 1 (Core) |
| abmb | None | None | Same as ctrl + melt anomaly | Medium | No | Tier 1 (Core) |
| exp01 | NorESM1-M | RCP8.5 | Open | Medium | No | Tier 1 (Core) |
| exp02 | MIROC-ESM-CHEM | RCP8.5 | Open | Medium | No | Tier 1 (Core) |
| exp03 | NorESM1-M | RCP2.6 | Open | Medium | No | Tier 1 (Core) |
| exp04 | CCSM4 | RCP8.5 | Open | Medium | No | Tier 1 (Core) |
| exp05 | NorESM1-M | RCP8.5 | Standard | Medium | No | Tier 1 (Core) |
| exp06 | MIROC-ESM-CHEM | RCP8.5 | Standard | Medium | No | Tier 1 (Core) |
| exp07 | NorESM1-M | RCP2.6 | Standard | Medium | No | Tier 1 (Core) |
| exp08 | CCSM4 | RCP8.5 | Standard | Medium | No | Tier 1 (Core) |
| exp09 | NorESM1-M | RCP8.5 | Standard | High | No | Tier 1 (Core) |
| exp10 | NorESM1-M | RCP8.5 | Standard | Low | No | Tier 1 (Core) |
| exp11 | CCSM4 | RCP8.5 | Open | Medium | Yes | Tier 1 (Core) |
| exp12 | CCSM4 | RCP8.5 | Standard | Medium | Yes | Tier 1 (Core) |
| exp13 | NorESM1-M | RCP8.5 | Standard | PIGL | No | Tier 1 (Core) |
| expA1 | HadGEM2-ES | RCP8.5 | Open | Medium | No | Tier 2 |
| expA2 | CSIRO-MK3 | RCP8.5 | Open | Medium | No | Tier 2 |
| expA3 | IPSL-CM5A-MR | RCP8.5 | Open | Medium | No | Tier 2 |
| expA4 | IPSL-CM5A-MR | RCP2.6 | Open | Medium | No | Tier 2 |
| expA5 | HadGEM2-ES | RCP8.5 | Standard | Medium | No | Tier 2 |
| expA6 | CSIRO-MK3 | RCP8.5 | Standard | Medium | No | Tier 2 |
| expA7 | IPSL-CM5A-MR | RCP8.5 | Standard | Medium | No | Tier 2 |
| expA8 | IPSL-CM5A-MR | RCP2.6 | Standard | Medium | No | Tier 2 |

## 2.3 Model set-up

Similar to the philosophy adopted for initMIP-Antarctica, there are no constraints on the method or datasets used to initialize ice sheet models. The exact initialization date is also left to the discretion of individual modeling groups, so the historical experiment length varies among groups (with some groups starting directly at the beginning of 2015 and therefore not sub-

mitting a historical run). The resulting ensemble includes a variety of model resolutions, stress balance approximations, and initialization methods, representative of the diversity of the ice sheet modeling community (see section 3 for more details on participating models).

The only constraints imposed on the ice sheet models are that they are able to simulate ice shelves and the evolution of grounding lines, as well as being able to use atmospheric and oceanic forcings varying in time and based on AOGCM outputs.



The inclusion of ice cliff failure, on the other hand, was not allowed, except in the ice shelf collapse experiments. Groups were invited to submit one or several sets of experiments, and modelers were asked to submit the full suite of open experiments (with the melt parameterization of their choice) and/or standard (Jourdain et al., under review) core experiments if possible. Unlike what was imposed for initMIP-Antarctica, models were free to include additional processes not specified here (e.g., changes in bedrock topography in response to changes in ice load or feedback between SMB and elevation).

Annual values for both scalar and two-dimensional outputs were reported on standard grids with resolutions of 4, 8, 16 or 32 km. Scalar quantities were recomputed from two-dimensional fields for consistency, and in order to create regional scalars used for the regional analysis. The two-dimensional fields were also regridded onto the standard 8-km grid, to facilitate spatial comparison and analysis. The requested outputs are listed in Appendix A. Each group also submitted a README file summarizing the model characteristics.

## 210  3  Participating models

16 sets of simulations from 13 groups were submitted to ISMIP6-Antarctica Projections. The groups and ice sheet modelers who ran the simulations are listed in table 2. Simulations are performed using various ice flow models, a range of grid resolutions, different approximations of the stress balance equation, varying basal sliding laws, multiple external forcings, and a diverse set of processes included in the simulations. Table 3 summarizes the main characteristics of the 16 simulations. Short
descriptions of the initialization method and main model characteristics are provided in Appendix C.

The 16 sets of submitted simulations have been performed using 10 different ice flow models. Amongst the simulations, 3 use the finite element method, 2 a combination of finite element and finite volume, and the remaining 11 the finite difference method. One simulation is based on a full-Stokes stress balance, two use the 3D Higher-Order approximations (HO, Pattyn, 2003), one is based on the L1L2 approximation (Hindmarsh, 2004), one on the shelfy-stream approximation (SSA, MacAyeal,
1989), while the other simulations combine the SSA with the shallow ice approximation (SIA, Hutter, 1982). The model resolutions range between 4 km and 20 km for models that use regular grids, but can be as low as 2 km in specific areas such as close to the grounding line or shear margins for models with spatially variable resolution (Morlighem et al., 2010).

As in initMIP-Antarctica (Seroussi et al., 2019), the initialization procedure reflects the broad diversity in the ice sheet modeling community: two simulations start from an equilibrium state, five models start from a long spin-up and three simulations
from data assimilation of recent observations. The remaining simulations combine the latter two approaches by either adding constraints to their spin-up (three simulations) or running short relaxations after performing data assimilation (three simulations). The initialization year varies between 1850 and 2015, so the length of the historical experiment varies between 0 and 115 years.

All submissions are required to include grounding line evolution (see section 2.3), but the treatment of grounding line
evolution and ocean melt in partially floating grid cells is left to the discretion of the modeling groups. Simulating ice front evolution (i.e., calving) in the simulations is also encouraged but not required, and the choice of ice front parameterization is free. Six models use a fixed ice front that does not involve in time (except for the ice shelf collapse experiments, for which





**Table 2.** List of participants, modeling groups and ice flow models in ISMIP6-Antarctica Projections

| Contributors | Group ID | Ice flow model | Group |
|---|---|---|---|
| Thomas Kleiner<br>Angelika Humbert | AWI | PISM | Alfred Wegener Institute for Polar and Marine Research,<br>Bremerhaven, Germany |
| Matthew Hoffman<br>Tong Zhang<br>Stephen Price | DOE | MALI | Los Alamos National Laboratory, Los Alamos, NM, USA |
| Ralf Greve<br><br>Reinhard Calov | ILTS_PIK | SICOPOLIS | Institute of Low Temperature Science,<br>Hokkaido University, Sapporo, Japan<br>Potsdam Institute for Climate Impact Research, Germany |
| Heiko Goelzer<br>Roderik van de Wal | IMAU | IMAUICE | Institute for Marine and Atmospheric Research,<br>Utrecht, The Netherlands |
| Nicole-Jeanne Schlegel<br>Hélène Seroussi | JPL | ISSM | Jet Propulsion Laboratory, California Institute of Technology, Pasadena, USA |
| Christophe Dumas<br>Aurelien Quiquet | LSCE | Grisli | Laboratoire des Sciences du Climat et de l'Environnement<br>Université Paris-Saclay, France |
| Gunter Leguy<br>William Lipscomb | NCAR | CISM | National Center for Atmospheric Research, Boulder, CO, USA |
| Ronja Reese<br>Torsten Albrecht<br>Ricarda Winkelmann | PIK | PISM | Potsdam Institute for Climate Impact Research, Germany |
| Tyler Pelle<br>Mathieu Morlighem<br>Hélène Seroussi | UCIJPL | ISSM | University of California, Irvine, USA<br><br>Jet Propulsion Laboratory, California Institute of Technology, Pasadena, USA |
| Frank Pattyn<br>Sainan Sun | ULB | f.ETISh | Université libre de Bruxelles, Belgium |
| Chen Zhao<br>Rupert Gladstone<br>Thomas Zwinger | UTAS | Elmer/Ice | University of Tasmania, Australia<br>Arctic Centre, University of Lapland, Finland<br>CSC IT Center for Science, Espoo, Finland |
| Jonas Van Breedam<br>Philippe Huybrechts | VUB | AISMPALEO | Vrije Universiteit Brussel, Belgium |
| Nicholas Golledge<br>Daniel Lowry | VUW | PISM | Antarctic Research Centre, Victoria University of Wellington,<br>and GNS Science, New Zealand |

retreat is imposed), while the other models rely on a combination of minimum ice thickness, strain rate values, and stress divergence to evolve the ice front position.

Ocean-induced melt rates under ice shelves follow the standard melt framework described in section 2.1.3 for 13 sets of simulations: 10 submissions use the non-local form, while 3 are based on the local form, and three of these 13 sets of simulations are based on the non-local or local anomaly forms (Jourdain et al., under review). The open melt framework was used by 8 sets of simulations that rely on a linear melt dependence of thermal forcing (Martin et al., 2011), a quadratic local melt



**Table 3.** List of ISMIP6-Antarctica Projections simulations and main model characteristics. Initialization methods used: Spin-up (SP), Spin-up with ice thickness target values (SP+, see Pollard and DeConto, 2012a), Data Assimilation (DA), Data Assimilation with relaxation (DA+), Data Assimilation of ice geometry only (DA*), and Equilibrium state (Eq). Melt in partially floating cells: Melt either applied or not over the entire cell based on a floating condition (Floating condition), N/A refers to models that do not have partially floating cells. Ice front migration schemes based on: strain rate (StR, Albrecht and Levermann, 2012), retreat only (RO), fixed front (Fix), minimum thickness height (MH) and divergence and accumulated damage (Div, Pollard et al., 2015). Basal melt rate parameterization in open framework: linear function of thermal forcing (Lin, Martin et al., 2011), quadratic local function of thermal forcing (Quad, DeConto and Pollard, 2016), PICO parameterization (PICO, Reese et al., 2018a), PICOP parameterization (PICOP, Pelle et al., 2019), plume model (Plume, Lazeroms et al., 2018), and Non-Local parameterization with slope dependence of the melt (Non-Local + Slope, Lipscomb et al., in prep.). Basal melt rate parameterization in standard framework: Local or Non-Local quadratic function of thermal forcing, Local or Non-Local anomalies (Jourdain et al., under review).

| Model name | Numerics | Stress balance | Resolution (km) | Init. Method | Initial Year | Melt in partially floating cells | Ice Front | Open melt parameterization | Standard melt parameterization |
|---|---|---|---|---|---|---|---|---|---|
| AWI_PISM | FD | Hybrid | 8 | Eq | 2005 | Sub-Grid | StR | Quad | Non-Local |
| DOE_MALI | FE/FV | HO | 2-20 | DA+ | 2015 | Floating condition | Fix | N/A | Non-Local anom. |
| ILTS_PIK_SICOPOLIS1 | FD | Hybrid | 8 | SP+ | 1990 | Floating condition | MH | N/A | Non-Local |
| IMAU_IMAUICE1 | FD | Hybrid | 32 | Eq | 1978 | No | Fix | N/A | Local anom. |
| IMAU_IMAUICE2 | FD | Hybrid | 32 | SP | 1978 | No | Fix | N/A | Local anom. |
| JPL1_ISSM | FE | SSA | 2-50 | DA | 2007 | Sub-Grid | Fix | N/A | Non-Local |
| LSCE_GRISLI | FD | Hybrid | 16 | SP+ | 1995 | N/A | MH | N/A | Non-Local |
| NCAR_CISM | FE/FV | L1L2 | 4 | SP+ | 1995 | Sub-Grid | RO | Non-Local + Slope | Non-Local |
| PIK_PISM1 | FD | Hybrid | 8 | SP | 1850 | Sub-Grid | StR | PICO | N/A |
| PIK_PISM2 | FD | Hybrid | 8 | SP | 2015 | Sub-Grid | StR | PICO | N/A |
| UCIJPL_ISSM | FE | HO | 3-50 | DA | 2007 | Sub-Grid | Fix | PICOP | Non-Local |
| ULB_FETISH_16km | FD | Hybrid | 16 | DA* | 2005 | N/A | Div | Plume | Non-Local |
| ULB_FETISH_32km | FD | Hybrid | 32 | DA* | 2005 | N/A | Div | Plume | Non-Local |
| UTAS_ElmerIce | FE | Stokes | 4-40 | DA | 2015 | Sub-Grid | Fix | N/A | Local |
| VUB_AISMPALEO | FD | SIA+SSA | 20 | SP | 2000 | N/A | MH | N/A | Non-Local anom. |
| VUW_PISM | FD | Hybrid | 16 | SP | 2015 | No | StR | Lin | N/A |

parameterization (DeConto and Pollard, 2016) but with a calibration different than the standard framework, a plume model
(Lazeroms et al., 2018), a box model (Reese et al., 2018a), a combination of box and plume models (Pelle et al., 2019)
or a non-local quadratic melt parameterization combined with ice shelf basal slope (Lipscomb et al., in prep.). Five sets of
simulations include results based on both the open and standard framework.

The modeling groups were asked to submit a full suite of core experiments based on the standard melt parameterization, the
open melt parameterization, or both. Most groups were able to do so, but several groups did not submit the ice shelf collapse
experiments, and one group (UTAS_ElmerIce) ran only a subset of experiments due to the high cost of running a full-Stokes
model of the entire Antarctic ice sheet. Simulations that initialize their model on January 2015 (see Table 3) do not have a



historical run, and their ctrl and ctrl_proj are identical. Seven submissions also performed some or all of the Tier 2 experiments based on the three additional AOGCM forcings. Table 4 lists all the experiments done by the modeling groups for both the core experiments and the Tier 2 experiments.

**Table 4.** List of experiments performed as part of ISMIP6-Antarctica Projections by the modeling groups.
* indicates simulations initialized directly at the beginning of 2015, for which ctrl and ctrl_proj experiments are identical.

| Experiment | AWI_PISM | DOE_MALI | ILTS_PIK_SICOPOLIS1 | IMAU_IMAUICE1 | IMAU_IMAUICE2 | JPL1_ISSM | LSCE_GRISLI | NCAR_CESM | PIK_PISM1 | PIK_PISM2 | UCIJPL_ISSM | ULB_fETISh_16 | ULB_fETISh_32 | UTAS_ElmerIce | VUB_AISMPALEO | VUW_PISM |
|---|---|---|---|---|---|---|---|---|---|---|---|---|---|---|---|---|
| historical | X | | X | X | X | X | X | X | X | | X | X | X | | X | X |
| ctrl | X | X | X | X | X | X | X | X | | | X | X | X | X | X | X |
| ctrl_proj | X | X* | X | X | X | X | X | X | X | X* | X | X | X | X* | X | X |
| asmb | X | X | X | X | X | X | X | X | X | X | X | X | X | | X | X |
| abmb | X | X | X | X | X | X | X | X | X | X | X | X | X | X | X | X |
| exp01 | X | | | | | | | X | X | X | X | X | X | | | X |
| exp02 | X | | | | | | | X | X | X | X | X | X | | | X |
| exp03 | X | | | | | | | X | X | X | X | X | X | | | X |
| exp04 | X | | | | | | | X | X | X | X | X | X | | | X |
| exp05 | X | X | X | X | X | X | X | X | | | X | X | X | X | X | |
| exp06 | X | X | X | X | X | X | X | X | | | X | X | X | X | X | |
| exp07 | X | X | X | X | X | X | X | X | | | X | X | X | | X | |
| exp08 | X | X | X | X | X | X | X | X | | | X | X | X | | X | |
| exp09 | X | X | X | X | X | X | X | X | | | X | X | X | | X | |
| exp10 | X | X | X | X | X | X | X | X | | | X | X | X | | X | |
| exp11 | X | | | | | | | | | | X | X | X | | | |
| exp12 | X | X | X | X | X | X | X | | | | X | X | X | | | |
| exp13 | X | X | X | X | X | X | X | X | | | X | X | X | X | X | |
| expA1 | X | | | | | | | X | | | | X | X | | | |
| expA2 | X | | | | | | | X | | | | X | X | | | |
| expA3 | X | | | | | | | X | | | | X | X | | | |
| expA4 | X | | | | | | | X | | | | X | X | | | |
| expA5 | X | X | | X | X | X | X | | | | X | X | X | | X | |
| expA6 | X | X | | X | X | X | X | | | | X | X | X | | X | |
| expA7 | X | X | | X | X | X | X | | | | X | X | X | | X | |
| expA8 | X | X | | X | X | X | X | | | | X | X | X | | | |



## 4   Results

We detail here the simulation results. We start by describing the initial state, as well as the historical and control runs. We then analyze the NorESM1-M RCP 8.5 runs, and the RCP 8.5 simulations based on different AOGCM forcing. Next, we compare the RCP 8.5 and RCP 2.6 results for the two AOGCMs selected to provide RCP 2.6 scenario forcings. We then investigate the effect of using the open and standard melt parameterizations. Finally, we explore the impact of uncertainties in ocean melt parameterization and the role of ice shelf collapse.

Results based on the open and standard melt parameterizations are combined, except in section 4.6 where we invesigate difference between these approaches. This means that 21 independent sets of results are extracted from the 16 submissions (8 based on the open melt framework and 13 based on the standard framework). No weighting based on number of submissions or agreement with observations is applied.

### 4.1   Historical run and 2015 conditions

As the initialization date for different models varies, all models run a short historical simulation until 2015. The length of this simulation varies between 165 years for PIK_PISM1, which starts in 1850, and 0 year for the three models (DOE_MALI, PIK_PISM2 and UTAS_ElmerIce) that start in 2015. During the historical run, simulations are forced with oceanic and atmospheric conditions representative of the conditions estimated during this period. The total annual SMB over Antarctica varies between 2200 and 3200 Gt/yr, with large interannual variations of up to 600 Gt/yr (see Fig. 1a). The total annual ocean induced basal melt rates under Antarctic ice shelves during the historical period varies between 0 and 2200 Gt/yr, with large interannual variations up to 1000 Gt/yr. The ice volume above floatation, however, experiences limited variations during the historical period, with less than 1000 Gt of change (Fig. 1b). The total ice mass above floatation varies between 1.99 and $2.15 \times 10^7$ Gt (between 54.9 and 59.3 m SLE) between the simulations, which is a 7% difference in the initial ice mass above floatation (Fig. 1c).

All historical simulations end in December 2014, at which point the projection experiments start. Figure 2 shows the total ice and floating ice extent for all submissions at the beginning of the experiments. The ice-covered area varies between 1.36 and $1.45 \times 10^7$ km$^2$, or 6.0%. There is good agreement between the modeled ice extent and the observed ice front (Howat et al., 2019) around the entire continent, which is a smaller spread compared to the initMIP-Antarctica submissions. The extent of ice shelves shown on Fig.2b varies between 1.19 and $1.89 \times 10^6$ km$^2$, or 29%, which is also reduced compared to the spread in initMIP-Antarctica, and in better agreement with observations (Rignot et al., 2011). Not only the large ice shelves, but also the smaller ice shelves of the Amundsen and Bellingshausen sea sectors, the Antarctic Peninsula, and Dronning Maud Land have a location and extent that is consistent with observations. A few models have ice shelves that extend slightly farther than the present-day ice over large parts of the continent, but they extend only a few tens of km past the observed ice front location. Finally, the location of the grounding line on the Ross ice streams fluctuates by several hundreds of km between the models, which is not surprising as the Ross ice streams rest over relatively flat bedrock, so small changes in model configuration lead to large variations in the grounding line position. The 2015 ice volume and ice volume above floatation are reported in table B1.





They indicate a variation of 6.8% of the total ice mass among the simulations, between 2.31 and 2.49 $\times 10^7$ Gt, and a variation of 7.7% in the total ice mass above floatation, between 1.99 and 2.15 $\times 10^7$ Gt or between 55.0 and 59.4 m of SLE, when the latest estimate is $57.9 \pm 0.9$ m (Morlighem et al., 2019a). Figure 3 shows the root mean square error (RMSE) between modeled and observed thickness and velocity at the beginning of the experiments. The RMSE thickness varies between 100 and 395 m, while the RMSE velocity varies between 90 and 440 m/yr and its logarithmic value between 0.79 and 2.2 log(m/yr), which is comparable to values reported for initMIP-Antarctica (Seroussi et al., 2019).

### 4.2 ctrl_proj

All the experiments start from the 2015 configuration and are run with varying atmospheric and oceanic forcings. The ctrl_proj experiment also starts from this configuration, but is run with constant climate conditions (no oceanic or atmospheric anomalies added), similar to those observed over the past several decades. The exact choice of forcing conditions for this run was not imposed and therefore varies between the simulations. Figure 1 shows that similarly to the historical run, the SMB and basal melt vary significantly between the simulations. The SMB varies between 2320 Gt/yr and 3090 Gt/yr, while the basal melt varies between 0 and 1750 Gt/yr. However, unlike what is observed in the historical run, there is no interannual fluctuation, since a mean climatology is used for this run.

During the 86 years of the ctrl_proj experiment, the evolution of ice mass above floatation varies between -50,000 and 47,000 Gt (between -130 and 140 mm SLE). As in initMIP-Antarctica, models initialized with a steady-state or a spin-up tend to have smaller trends than models initialized with data assimilation. The trend in the ctrl_proj mass above floatation is significant in several models and negligible in others. Since constant climate conditions are applied, trends cannot be considered as a physical response of the Antarctic ice sheet, but rather highlight the impact of model choices to initialize the simulation and represent ice sheet evolution, the lack of physical processes (Pattyn, 2017), the limited number or inaccuracy of observations (Seroussi et al., 2011; Gillet-Chaulet et al., 2012), and the need to better integrate observations in ice flow models (Goldberg et al., 2015; Nowicki and Seroussi, 2018).

All the results presented in the remainder of the manuscript are shown relative to the outputs from the ctrl_proj experiment. As a consequence, these results should be interpreted as the response to additional climate change compared to a scenario where the climate remains constant and similar to the past few decades. Submissions that include both open and standard experiment results can have significant variations in their historical and ctrl_proj depending on whether the open or standard melt parameterization is used (see Fig. 1). Outputs from the ctrl_proj run the open or standard melt parameterization are therefore respectively removed from the experiments based on the open or standard framework when possible.

### 4.3 NorESM1-M RCP 8.5 scenario

The NorESM1-M RCP 8.5 scenario produces mid-to-high changes in the ocean and low changes in the atmosphere over the 21[st] century compared to other CMIP5 AOGCMs (Barthel et al., in review). The impacts of these changes on the evolution of the Antarctic ice sheet are summarized in Fig. 4, 5, and 6. Figure 4 shows that under this forcing, the Antarctic ice sheet loses a volume above floatation varying between -26 and 226 mm of SLE between 2015 and 2100, relative to ctrl_proj experiments.



The impact of the forcing remains limited until 2050, with changes less than $\pm$ 25 mm. It quickly increases after 2050, at which point the simulations start to diverge strongly.

Figure 5 shows that the sea level contribution and the mechanisms at play vary significantly for the West Antarctic Ice Sheet (WAIS), East Antarctic Ice Sheet (EAIS) and the Antarctic Peninsula. In WAIS, the additional SMB is limited to a few millimeters (between -4 and 2 mm SLE), and all models predict a mass loss varying between 0 and 157 mm SLE relative to ctrl_proj. EAIS experiences a significant increase in SMB, with a cumulative additional SMB causing between 17 and 48 mm SLE of mass gain relative to ctrl_proj. This mass gain is partially offset by the dynamic response of outlet glaciers in EAIS, resulting in a total volume change varying between a 25 mm SLE mass gain and 168 mm SLE mass loss. The small size of the Antarctic Peninsula and limited mass of its glaciers make it a smaller contributor to sea level change compared to WAIS and EAIS: the contribution to sea level varies between -5 and 8 mm SLE relative to ctrl_proj, with a signal slit by the additional SMB (between 1 and 3 mm SLE mass gain) and dynamic response . These results therefore highlight the contrast between the EAIS and Antarctic Peninsula, which are projected to either gain or lose mass and where SMB changes are relatively large, and the WAIS, which is dominated by a dynamic mass loss caused by the changing ocean conditions.

Regions with the largest changes can also be seen in figure 6, which shows the mean change in thickness and velocity between 2015 and 2100 for the 21 NorESM1-M simulations relative to ctrl_proj. Most Antarctic ice shelves thin by 10 m or more over the 86-year simulation, with the Ross ice shelf experiencing the largest thinning of 50 m on average (Fig. 6a). This thinning does not propagate to the ice streams feeding the ice shelves, except for Thwaites Glacier in the Amundsen Sea Sector and Totten Glacier in Wilkes Land. Many coastline regions, on the other hand, experience small thickening, as is the case for the Antarctic Peninsula, Dronning Maud Land and Kamp Land, where the relative thickening is about 3 m. Variations between the simulation are large and dominate the signal in many places (Fig. 6c). Changes in velocity (Fig. 6b) over ice shelves are more limited and are not homogeneous, with acceleration close to the grounding line areas and slowdown close to the ice front, as observed for the Ross and Ronne-Filchner ice shelves. Some acceleration is observed on grounded parts of Thwaites, Pine Island and Totten Glaciers as well. However, there is a large discrepancy in velocity changes among the simulations, and the standard deviation in velocity change in larger than the mean signal over most of the continent (Fig. 6d).

## 4.4 RCP 8.5 scenario: impact of AOGCMs

Outputs from six CMIP5 AOGCMs were used to perform RCP 8.5 experiments (see Table 1). Figure 7 shows the evolution of the ice volume above floatation relative to ctrl_proj for all the individual RCP 8.5 simulations performed, as well as the mean values for each AOGCM. As seen above for NorESM1-M, changes are small for most simulations until 2050, after which differences between AOGCMs and ice flow simulations start to emerge. Runs with HadGEM2-ES lead to significant sea level rise, with a mean ice mass loss of 101 mm SLE (standard deviation 75 mm SLE) for the 15 submissions of expA1 and expA5. Runs performed with CCSM4 show the largest ice mass gain, with a mean gain of 32 mm SLE (standard deviation 50 mm SLE) for the 21 submissions of exp04 and exp08. Results for CSIRO-MK3 and IPSL-CM5A-MR are similar to CCSM4, but with slightly lower mass gain on average, while results from MIROC-ESM-CHEM are similar to NorESM1-M.





Figure 8 shows the regional differences in these contributions relative to ctrl_proj. WAIS loses mass with three of the
AOGCMs, gains mass with CSIRO-MK3, while its contribution is uncertain with CCSM4 and IPSL-CM5A-MR. For the
EAIS, results from 5 out of 6 AOGCMs lead to a clear mass gain. Only HadGEM2-ES forcing causes a mass loss in EAIS,
with $25 \pm 27$ mm SLE. Uncertainties for the WAIS are larger than for the EAIS, as the ocean plays a significant role in this
region. As observed in initMIP-Antarctica (Seroussi et al., 2019), changes in oceanic conditions lead to a much larger spread
in ice sheet evolution than changes in SMB, even with simplified forcing. Changes in the Antarctic Peninsula lead to mass
change between -9 and 15 mm SLE.

### 4.5  Impact of scenario: RCP 8.5 and RCP 2.6

Two AOGCMs were chosen to run both RCP 8.5 and RCP 2.6 experiments: NorESM1-M and IPSL-CM5A-MR. Figure 9
shows the evolution of the Antarctic ice sheet under these two scenarios relative to ctrl_proj for both AOGCMs. Only ice flow
models that performed both RCP 8.5 and RCP 2.6 experiments were used to compare these scenarios, so two RCP 8.5 runs
were not included, leading to the analysis of 19 NorESM1-M and 14 IPSL-CM5A-MR pairs of experiments.

Results from NorESM show no significant change between the two scenarios in terms of ice volume above floatation by
2100 (Fig. 9a). Both scenarios lead to a mean sea level contribution of about 16 mm SLE in 2100, with a higher standard
deviation for the RCP 8.5 scenario (39 mm for RCP 8.5 and 30 mm for RCP 2.6). However, the overall similar behavior hides
large regional differences revealed in figure 10a. The WAIS loses more mass in RCP 8.5 compared to RCP 2.6, while the EAIS
gains more ice mass. The additional SMB is larger for all regions under RCP 8.5 (20 mm SLE in the EAIS and 2 mm SLE for
the the Peninsula), but is compensated by a large dynamic response to ocean changes in the WAIS.

Simulations based on IPSL-CM5A-MR, on the other hand, show significant differences in ice contribution to sea level at a
continental scale. Ice contributes to $-17 \pm 13$ mm SLE for the RCP 8.5 scenario and $0 \pm 5$ mm SLE for the RCP 2.6 scenario
(Fig. 9). For RCP 2.6, the overall mass loss in the WAIS is compensated by mass gain in the EAIS, leading to an overall mass
that is nearly constant (Fig. 10). For RCP 8.5, on the other hand, there are large mass gains in all ice sheet regions as SMB
increases significantly. Only a few simulations show mass loss of the WAIS relative to ctrl_proj. Similar to what is observed
for NorESM1-M, the uncertainty is large for RCP 8.5, as oceanic changes are more pronounced in this scenario.

Overall, these two AOGCMs respond very differently to increased carbon concentrations, which is reflected in the differences
in ice sheet evolution.

### 4.6  Impact of open vs standard melt framework

All of the RCP 8.5 and RCP 2.6 experiments were simulated with both open and standard melt frameworks. The standard
framework allows us to assess the uncertainty associated with ice flow models when the processes controlling ice–ocean
interactions are fixed. The open framework, in contrast, allows for additional uncertainties due to the physics of ice–ocean
interactions. We now investigate the impacts of these different approaches on simulation results.
Figure 11 shows the cumulative ocean-induced basal melt and the change in ice volume above floatation between 2015 and
2100 and relative to ctrl_proj, for the six RCP 8.5 experiments and for the 8 and 14 submissions using the open and standard



melt frameworks, respectively. The basal melt applied in the standard framework is higher than the basal melt resulting from the open framework for about half of the experiments and Antarctic regions and lower for the other half. The standard deviation of basal melt is larger in the open melt framework (see Fig. 11a), which is expected given the additional flexibility in the melt

parameterization and the wide range of melt parameterizations used in the open framework (see Table 3). However, despite the similar melt rates applied, the sea level contribution relative to ctrl_proj is higher (either more mass loss or less mass gain) in the open framework than in the standard framework, regardless of the region and the AOGCM. The mean additional sea level contribution (either more mass loss or less mass gain) simulated in the open framework is 28 mm SLE for WAIS and 27 mm for EAIS.

### 4.7 Impact of melt uncertainties


The impact of melt uncertainties is assessed exclusively for the standard melt parameterization framework, for which different choices of parameters can be used in a similar way by all models. Here we assess the impact of two sources of uncertainty that impact the choice of $\gamma_0$ and the regional $\delta_T$ values. The melt parameterization provides a distribution of $\gamma_0$, and the median value is used for most experiments (see table 1). Two experiments (exp09 and exp10) use the $5^{\text{th}}$ and $95^{\text{th}}$ percentile

values of the distribution to estimate the impact of parameter uncertainty on basal melt and ice mass loss. A third experiment investigates the impact of the dataset used to calibrate the melt parameterization (exp13): instead of using all the melt rates and ocean conditions around Antarctica, it uses only the high melt values near the Pine Island ice shelf grounding line ("PIGL" coefficient, see section 2.1.3), which results in $\gamma_0$ an order of magnitude higher. All these experiments are based on NorESM1-M and RCP 8.5, so the applied SMB is similar in all experiments; only the basal melt differs. The initial basal melt is calibrated

to be equal to observed values (Rignot et al., 2013; Depoorter et al., 2013) in each case and for each Antarctic basin, so only the initial distribution of melt and its evolution in time vary, not its total initial magnitude.

Fig.12a shows the impact of using the $5^{\text{th}}$, $50^{\text{th}}$, and $95^{\text{th}}$ percentile values of the $\gamma_0$ distribution for models that performed these three experiments. The total melt starts from similar values but diverges quickly as ocean conditions change. By 2100, the mean total melt applied is 3,100 Gt/yr for the median value, while it is 2,700 Gt/yr and 3600 Gt/yr respectively for the $5^{\text{th}}$

and $95^{\text{th}}$ percentile values of the $\gamma_0$ distribution. While these differences represent about 15% of the total melt applied, they fall largely within the spread of basal melt values applied for the median $\gamma_0$ for the different simulations and are smaller than interannual variations. Impacts of these changes on ice dynamics are shown on Fig.12c. The mean sea level contributions with the median $\gamma_0$ is 1.9 mm SLE, while it is -0.4 and 4.0 mm SLE 2100 for the $5^{\text{th}}$ and $95^{\text{th}}$ percentile. The overall evolution of Antarctica remains similar until about 2030, at which point the three experiments start to diverge.

Fig.12 also highlights the role of the calibration datasets. The "MeanAnt" and "PIGL" experiments start with similar total melt values and are both calibrated to be in agreement with current observations of melt (because models have initial geometries that differ from observations, they have minor differences in the amount of total initial melt). The total melt diverges between the two experiments after just a few years, and continues to diverge during the $21_{\text{st}}$ century as ocean conditions and ice shelf configurations change, reaching 3,100 and 6,900 Gt/yr on average in 2100 for the "MeanAnt" and "PIGL" experiments

(Fig.12b), respectively. The impact on ice dynamics and sea level is large, with six times larger mean contribution to sea level





by 2100 relative to ctrl_proj for the "PIGL" experiment, reaching a mean SLE contribution of 32 mm, see Fig.12d). This is the simulation with the greatest amounts of ice loss, with models predicting mass loss of up to 30 cm SLE by 2100. This melt parameterization causes larger melt rates close to grounding lines and higher sensitivity, as $\gamma_0$ is an order of magnitude larger for this "PIGL" parameterization than for the "MeanAnt" parameterization. This run thus represents an upper end to plausible

values for sub-shelf melting, yet it is calibrated to simulate initial basal melting in agreement with present-day observations. It also highlights the non-linear ice sheet response to submarine melt forcing: the doubling of in basal melt leads to more than ten times greater ice mass loss.

## 4.8 Impact of ice shelf collapse

The impact of ice shelf collapse is tested with exp11 and exp12 for the open and standard frameworks, respectively. These

experiments are based on outputs from CCSM4 and are similar to exp04 and exp08: the SMB and ocean thermal forcing are similar, so the two sets of experiments only differ by the inclusion of ice shelf collapse. As mentioned in section 2.1.4, the processes included in the response of the tributary ice streams feeding into these ice shelves is left to the judgement of modeling groups. However, no group included the marine ice cliff instability (Pollard et al., 2015) following ice shelf collapse. Only the 14 simulations (including 4 open and 10 standard melt parameterizations) that performed the ice shelf collapse experiments are

included in the following figures. Results from 7 simulations of exp04 and exp08 were therefore excluded from the ensemble with no ice shelf collapse.

As shown in Nowicki et al. (in review), the presence of significant liquid water on the surface of ice shelves is modeled for less than 60,000 km$^2$ until 2050, so ice shelf collapse is limited. Starting in 2050, it rapidly increases, reaching 450,000 km$^2$ by 2100. The evolution of ice shelf extent in the ice sheet simulations reflects this evolution: Figure 13a, shows the evolution

of ice shelf extent for the CCSM4 simulations with and without ice shelf collapse. As the external forcings are similar in both runs, the difference comes from the ice shelf collapse and the response to this collapse. In the simulations without collapse, ice shelf extent remains relatively constant, with less than 40,000 km$^2$ change on average compared to ctrl_proj. When ice shelf collapse is included, ice shelf extent is reduced by an average of 360,000 km$^2$ between 2015 and 2100 compared to the ctrl_proj runs.

While ice shelf collapse does not directly contribute to sea level rise, the dynamic response of the ice streams to the colapse leads to an average of 8 mm SLE difference between the two scenarios (Fig. 13a). These changes occur largely over the Antarctic Peninsula, next to George V ice shelf, but also on Totten Glacier (see Fig.14a). Including ice shelf collapse also leads to an acceleration of up to 100 m/yr in these same regions (see Fig.14b). Large uncertainties dominate these model responses, however.

The ice shelf collapse experiments are based on CCSM4, as this model shows the largest potential for ice shelf collapse out of the six AOGCMs selected (Nowicki et al., in review). Similar experiments performed with other AOGCMs are therefore expected to show a lower impact of ice shelf collapse.



## 5   Discusssion

ISMIP6-Antarctica Projections under the RCP 8.5 scenario show a large spread of Antarctic ice sheet evolution over 2015–
2100, depending on the ice flow model adopted, the AOGCM forcings applied, the ice sheet model processes included, and the
form and calibration of the basal melt parametrization. The Antarctic contribution to sea level with the "MeanAnt" calibration in
response to this scenario varies between a sea level drop of 7.8 cm and a sea level increase of over 28 cm, compared to a constant
climate similar to that of the past few decades. Contributions up to 30 cm are also simulated when the melt parameterization is
calibrated with high melt rates in Pine Island cavities (see section 4.7). Such a parameterization is also calibrated with present-
day observations but has a much stronger sensitivity to ocean forcing (Jourdain et al., under review), leading to more rapid
increases in basal melting as ocean waters in ice shelf cavities warm. As observations of ocean conditions within ice shelf
cavities and the resulting ice shelf melt rates remain limited, these numbers cannot be excluded from consideration.

All the numbers reported here describe Antarctic mass loss relative to that from a constant climate, so the mass loss trend
over the past few decades needs to be added to obtain a total Antarctic contribution to sea level through 2100. The recent
IMBIE assessment estimated the Antarctic mass loss between 38 and 219 Gt/yr, depending on the time period considered
(Shepherd et al., 2018), which corresponds to a cumulative mass loss of 9 and 52 mm over 2015–2100. Adding this to the
range of Antarctic mass loss simulated as part of ISMIP6 gives a range of between -6.9 and 35 cm SLE. These numbers cover
the wide range of results previously published (e.g., Edwards et al., 2019; DeConto and Pollard, 2016; Schlegel et al., 2018;
Golledge et al., 2019) but don't allow to reproduce the highest contributions up to 1 meter previously reported. These numbers
show less spread than the simulations performed under the SeaRISE experiments, mostly due to the lower basal melt anomalies
applied under ice shelves (Bindschadler et al., 2013; Nowicki et al., 2013a). They are also similar to numbers presented in the
Pachauri et al. (2014): the likely range (5–95% of model range) of Antarctic contribution to global-mean sea-level rise between
the 1986-2005 period and 2100 under RPC 8.5 scenario was between -8 and 14 cm.

The response of the ice sheet changes in ocean forcings varies significantly spatially, suggesting that some sectors of the
ice sheet are significantly more vulnerable to changes in ocean circulation than others. Figure 15 shows the sensitivity of the
18 Antarctic basins (Rignot et al., 2019) to changes in oceanic conditions for all RCP 8.5 experiments; the dynamic mass loss
(total ice above floatation mass loss minus SMB change) between 2015 and 2100 is represented as a function of the cumulative
ocean induced melt over the same period, both relative to ctrl_proj. The Amundsen Sea sector and Wilkes Land show the largest
sensitivity to changes in oceanic conditions. Glaciers feeding the West Side of the Ross ice shelf show the smallest response
to increased basal melt, followed by the Ross ice streams and glaciers feeding the Ronne ice shelf. For the other regions, none
of the simulations predicted large increase in oceanic induced melt by 2100 so we cannot conclude on the sensitivity of these
sectors to oceanic forcings.

The large spread in Antarctic ice sheet projections reported here contrasts with the relatively narrow range of projections
reported in Goelzer et al. (sub.) for the Greenland ice sheet. We attribute this difference to the dominant role of SMB in driving
future evolution of Greenland and the more constrained forcing applied for ice front retreat in Greenland.





Uncertainties in the sea level estimates come from the spread in AOGCM forcing (see section 4.4), the melt parameterization adopted and its calibration (see sections 4.6 and 4.7), and the spread caused by the choices made by the ice flow models (see section 4.3 and Seroussi et al. (2019)). All these sources of uncertainty impact the results, and uncertainties in ocean conditions and their conversion into basal melt rates through parameterization lead to the largest spread of results, especially when different

datasets are used for parameter calibration. Antarctic mass losses above 20 cm SLR by 2100 are reached only with the PIGL calibration (Fig. 12) or the open melt framework. Furthermore, not only does the magnitude of basal melt influence Antarctic dynamics, but the spatial distribution of melt rates has a strong impact on the results, as observed when comparing the open and standard experiments (4.6). These findings are similar to those described by Gagliardini et al. (2010) based on idealized model configurations and highlight the need to use coupled ice-ocean models to better understand ice-ocean interactions and

represent them in ice flow models (Seroussi et al., 2017; Favier et al., 2019).

The results presented here do not include any weighting of the ice flow models based on their agreement with observations or the number of simulations submitted. As explained in previous studies (Goelzer et al., 2017, 2018; Seroussi et al., 2019), the range of initialization techniques adopted by models leads to varying biases. Some models are initialized with a long paleoclimate spin-up, giving limited spurious trends but an initial configuration further from the observed state, whereas models

initialized with data assimilation of present-day observations can capture these conditions accurately but often have non-physical trend in their evolution. Assigning weights to different models is therefore a complicated question that is not addressed in the present study, but that might lead to an overrepresentation of the models that submitted several contributions. The approach taken here (i.e., no weighting) is similar to that adopted within the larger CMIP framework.

The simulations performed as part of ISMIP6-Antarctica Projections represent a significant improvement compared to pre-

vious intercomparisons of Antarctic evolution, especially in terms of the treatment of ice shelves, grounding line evolution, and ocean-induced basal melt (Bindschadler et al., 2013; Nowicki et al., 2013a). These are representative of improvements made to ice flow models over the past decade (Pattyn et al., 2018). However, several limitations remain, regarding both external forcings (Nowicki and Seroussi, 2018) and ice flow models (Pattyn et al., 2018). SMB forcing from AOGCMs generally has a coarse resolution, and no regional model was used to downscale the forcing, unlike what was done for Greenland (Nowicki

et al., in review; Goelzer et al., sub.), so SMB in regions with steep surface slopes might not be well captured. The inclusion of surface-elevation feedbacks (Helsen et al., 2012) was left to the discretion of ice modeling groups, and no models included one, so this positive feedback was neglected in the present simulations. Because CMIP5 AOGCMs do not include ocean circulation under ice shelves, several simplifying assumptions must be made to estimate ocean conditions in ice shelf cavities (Jourdain et al., under review). Ice–ocean interactions in ice shelf cavities are poorly observed and constrained (Dutrieux et al., 2014;

Jenkins et al., 2018; Holland et al., 2019), leading to additional limitations on the representation of ocean- induced sub-shelf melt. Finally, despite the progresses in ice sheet numerical modeling over the last decade (Pattyn et al., 2018; Goelzer et al., 2017), significant limitations remain in our understanding of basal sliding (Brondex et al., 2019), basal hydrology (De Fleurian et al., J. Glaciol.), calving (Benn et al., 2017) or interaction with Solid Earth (Gomez et al., 2015; Larour et al., 2019).

The analysis of the simulations conducted here is presented relative to the ctrl_proj, and current trends in Antarctic mass

loss are added afterwards. It was decided that using results of ice flow simulations directly, without subtracting the trend from





a control run, is not yet appropriate given the large trend in the historical simulations and ctrl experiments (Fig. 1). Such a trend does not represent recent physical changes but rather limitations in observations (Seroussi et al., 2011), external forcings (Nowicki and Seroussi, 2018), ice flow models (Pattyn et al., 2018), and procedures used to initialize ice flow models (Seroussi et al., 2019; Nowicki and Seroussi, 2018; Goldberg et al., 2015). As ice sheets respond non-linearly to changes, such an

approach introduces a bias in the ice response, but these approach was deemed to be the most appropriate approach given current limitations. This same approach has been adopted in other recent ice flow modeling studies (Nowicki et al., 2013a, b; Schlegel et al., 2018). The choice of AOGCMs was made to cover a large range of responses to RCP scenarios, but is not representative of the mean changes exhibited by CMIP5 AOGCMs (Barthel et al., in review). As a result, we expect that the spread of model response represented here covers the diversity of AOGCM outputs. However, computing mean values using

different AOGCMs should be avoided, as only a few AOGCMs were sampled. Finally, all the results presented here are based on CMIP5 AOGCMs. Additional results based on CMIP6 AOGCMs will be presented in following publications.

## 6    Conclusions

We present here simulations of the Antarctic ice sheet evolution between 2015 and 2100 from a multi-model ensemble, as part of the ISMIP6 framework. Ice sheet models from 15 international ice sheet modeling groups are forced with outputs from

AOGCMs chosen to represent a large spread of possible evolution of oceanic and atmospheric around Antarctica over the $21^{st}$ century. Results show that the Antarctic ice sheet will contribute between -7.8 and 30.0 cm of SLE under RCP 8.5 scenario compared to an ice sheet forced under constant conditions representative of the past decade. AOGCMs suggest significant increase in SMB that are partially balanced by dynamic changes in response to ocean warming. Strong regional differences exist: WAIS loses mass under most scenarios and for all models, as the increase in SMB remains limited but the increase

in ice discharge are large. EAIS, on the other hand, gains mass in many simulations, as dynamic mass loss is too limited to compensate the large increase in SMB. The evolution of the Antarctic ice sheet under the RCP 2.6 scenario has a similar behavior, but with a smaller spread of SLE contribution between -1.4 and 17.7 cm relative to a constant forcing, with less SMB increase and a smaller dynamic response. The main sources of uncertainties remain the physics of ice flow models and the representation of ocean-induced melt at the base of ice shelves.

*Data availability.*  Model outputs from the simulations described in this paper will be made available in the CMIP6 archive through the Earth System Grid Federation (ESGF) with digital object identifier https://doi.org/xxx. In order to document CMIP6's scientific impact and enable ongoing support of CMIP, users are obligated to acknowledge CMIP6, participating modeling groups, and the ESGF centres (see details on the CMIP Panel website at http://www.wcrpclimate.org/index.php/wgcm-cmip/about-cmip). The forcing datasets are available through the ISMIP6 wiki and are also made publicly available via https://doi.org/xxx.





**Table A1.** Data requests for Antarctica-Projections. ST: State variable, FX: Flux variable, CST: Constant

| Variable name | Type | Standard name | Unit |
|---|---|---|---|
| Ice sheet thickness | ST | land_ice_thickness | m |
| Ice sheet surface elevation | ST | surface_altitude | m |
| Ice sheet base elevation | ST | base_altitude | m |
| Bedrock elevation | ST | bedrock_altitude | m |
| Geothermal heat flux | CST | upward_geothermal_heat_flux_at_ground_level | $W\,m^{-2}$ |
| Surface mass balance flux | FL | land_ice_surface_specific_mass_balance_flux | $kg\,m^{-2}\,s^{-1}$ |
| Basal mass balance flux | FL | land_ice_basal_specific_mass_balance_flux | $kg\,m^{-2}\,s^{-1}$ |
| Ice thickness imbalance | FL | tendency_of_land_ice_thickness | $m\,s^{-1}$ |
| Surface velocity in x direction | ST | land_ice_surface_x_velocity | $m\,s^{-1}$ |
| Surface velocity in y direction | ST | land_ice_surface_y_velocity | $m\,s^{-1}$ |
| Surface velocity in z direction | ST | land_ice_surface_upward_velocity | $m\,s^{-1}$ |
| Basal velocity in x direction | ST | land_ice_basal_x_velocity | $m\,s^{-1}$ |
| Basal velocity in y direction | ST | land_ice_basal_y_velocity | $m\,s^{-1}$ |
| Basal velocity in z direction | ST | land_ice_basal_upward_velocity | $m\,s^{-1}$ |
| Mean velocity in x direction | ST | land_ice_vertical_mean_x_velocity | $m\,s^{-1}$ |
| Mean velocity in y direction | ST | land_ice_vertical_mean_y_velocity | $m\,s^{-1}$ |
| Ice surface temperature | ST | temperature_at_ground_level_in_snow_or_firn | K |
| Ice basal temperature | ST | land_ice_basal_temperature | K |
| Magnitude of basal drag | ST | magnitude_of_land_ice_basal_drag | Pa |
| Land ice calving flux | FL | land_ice_specific_mass_flux_due_to_calving | $kg\,m^{-2}\,s^{-1}$ |
| Grounding line flux | FL | land_ice_specific_mass_flux_due_at_grounding_line | $kg\,m^{-2}\,s^{-1}$ |
| Land ice area fraction | ST | land_ice_area_fraction | 1 |
| Grounded ice sheet area fraction | ST | grounded_ice_sheet_area_fraction | 1 |
| Floating ice sheet area fraction | ST | floating_ice_sheet_area_fraction | 1 |
| Total ice sheet mass | ST | land_ice_mass | kg |
| Total ice sheet mass above floatation | ST | land_ice_mass_not_displacing_sea_water | kg |
| Area covered by grounded ice | ST | grounded_land_ice_area | $m^2$ |
| Area covered by floating ice | ST | floating_ice_shelf_area | $m^2$ |
| Total SMB flux | FL | tendency_of_land_ice_mass_due_to_surface_mass_balance | $kg\,s^{-1}$ |
| Total BMB flux | FL | tendency_of_land_ice_mass_due_to_basal_mass_balance | $kg\,s^{-1}$ |
| Total calving flux | FL | tendency_of_land_ice_mass_due_to_calving | $kg\,s^{-1}$ |
| Total grounding line flux | FL | tendency_of_grounded_ice_mass | $kg\,s^{-1}$ |

**Appendix A: Requested outputs**

The model outputs requested as part of ISMIP6 are listed in Table A1. Annual values were submitted for both scalar and two-dimensional variables. Flux variables reported are averaged over calendar years, while state variables are reported at the end of calendar years.



**Table B1.** Simulated Antarctic ice mass, ice mass above floatation, total ice extent and floating ice extent at the beginning of the experiments

| Model name | Ice Mass (10⁷ Gt) | Ice Mass Above Floatation (10⁷ Gt) | Total ice extent (10⁷ km²) | Floating ice extent (10⁶ km²) |
|---|---|---|---|---|
| AWI_PISM_std | 2.49 | 2.14 | 1.43 | 1.25 |
| AWI_PISM_open | 2.49 | 2.14 | 1.43 | 1.25 |
| DOE_MALI_std | 2.44 | 2.10 | 1.38 | 1.47 |
| ILTS_PIK_SICOPOLIS1_std | 2.45 | 2.12 | 1.40 | 1.64 |
| IMAU_IMAUICE1_std | 2.32 | 1.99 | 1.41 | 1.51 |
| IMAU_IMAUICE2_std | 2.31 | 1.99 | 1.41 | 1.52 |
| JPL1_ISSM_std | 2.44 | 2.10 | 1.39 | 1.45 |
| LSCE_GRISLI_std | 2.47 | 2.13 | 1.40 | 1.46 |
| NCAR_CISM_std | 2.41 | 2.08 | 1.38 | 1.30 |
| NCAR_CISM_open | 2.41 | 2.08 | 1.38 | 1.30 |
| PIK_PISM1_open | 2.48 | 2.15 | 1.38 | 1.43 |
| PIK_PISM2_open | 2.49 | 2.15 | 1.39 | 1.44 |
| UCIJPL_ISSM_std | 2.40 | 2.08 | 1.36 | 1.47 |
| UCIJPL_ISSM_open | 2.40 | 2.08 | 1.36 | 1.47 |
| ULB_fETISh_16_std | 2.42 | 2.07 | 1.45 | 1.92 |
| ULB_fETISh_16_open | 2.42 | 2.07 | 1.45 | 1.89 |
| ULB_fETISh_32_std | 2.43 | 2.08 | 1.42 | 1.70 |
| ULB_fETISh_32_open | 2.43 | 2.08 | 1.41 | 1.63 |
| UTAS_ELmerIce_std | 2.43 | 2.09 | 1.41 | 1.35 |
| VUB_AISMPALEO_std | 2.49 | 2.14 | 1.42 | 1.19 |
| VUW_PISM_open | 2.43 | 2.07 | 1.39 | 1.34 |

## Appendix B: Initial Values

We report here the scalar values of simulated Antarctic ice sheet ice mass, ice mass above floatation, ice extent, and ice shelf extent in Table B1. Values are reported at the beginning of January 2015, when the experiments start.

## Appendix C: Ice flow model initialization and characteristics

The descriptions below summarize the initialization procedure and main characteristics by the different ice flow modeling groups.

**AWI_PISM**

The AWI_PISM ice sheet model is based on the Parallel Ice Sheet Model (PISM, Bueler and Brown, 2009; Winkelmann et al., 2011; Aschwanden et al., 2012) version 1.1.4 with modifications for ISMIP6. PISM solves a hybrid combination of the non-





sliding shallow ice approximation (SIA) and the shallow shelf approximation (SSA) for grounded ice, where the SSA solution
acts as a sliding law, and only the SSA for floating ice. PISM also solves for Enthalpy to account for the temperature and
water content of the ice in the rheology. The model uses a structured rectangular grid with a uniform horizontal resolution of
8 km (16 km early in the spin-up) and 81 vertical z–coordinate levels that are refined towards the base. The total ice domain
height is 6000 m with an additional heat conducting bedrock layer of 2000 m thickness (21 equal levels). The calving front
can evolve freely on sub-grid scale (Albrecht et al., 2011). In addition to calving below a certain thickness threshold (here
150 m), a kinematic first-order calving law, called Eigen-calving (Levermann et al., 2012), is utilized with the calving parameter
$K = 10^{17}$ m s. Floating ice that extends far into the open ocean (seafloor elevation reaches 2000 m below sea level) is also
calved off. The grounding line position is determined using hydrostatic equilibrium. Basal friction in partially grounded cells is
weighted according to the grounded area fraction (Feldmann et al., 2014). The non-local quadratic melt scheme and the related
data sets provided by ISMIP6 are used to compute the ice shelf basal melt in the spin-up and all "standard" experiments. For
the "open" experiments, the local quadratic melt scheme is used. Ice shelf basal melt is applied on sub-grid scale.

To initialize the model, an equilibrium-type spin-up based on steady present-day climate has been performed. Atmo-
spheric forcing (2m air temperature and precipitation) is the multi-annual mean 1995–2014 (ISMIP6 reference period) from
RACMO2.3p2 (van Wessem et al., 2018). For the surface mass balance, a positive degree-day scheme (Huybrechts and
de Wolde, 1999; Martin et al., 2011) is used. Geothermal heat flux is from (Shapiro and Ritzwoller, 2004) and the bedrock
elevation is fixed in time. The ocean is forced with the present-day ocean forcing field provided by ISMIP6. The spin-up con-
sists of an initialization with idealized temperature-depth profiles, a 100-year geometry relaxation run and a 200 kyrs thermo-
mechanically coupled run with fixed geometry for thermal equilibration. For those stages, the non-sliding SIA is used on a
16 km horizontal grid. After re-gridding the output (except the geometry) onto the final 8 km grid, the model runs for 30 kyrs
using full model physics and a freely evolving geometry. The initial ice sheet geometry for the spin-up is based on Bedmap2
(Fretwell et al., 2013) and is refined in the Recovery Glacier area with additional ice thickness data sets (Humbert et al., 2018;
Forsberg et al., 2018). The historical simulation from January 2005 until end of December 2014 employs the NorESM1-M-
RCP8.5 atmospheric and oceanic forcing.

**DOE_MALI**

MPAS-Albany Land Ice (MALI) (Hoffman et al., 2018) uses a three-dimensional, first-order "Blatter-Pattyn" momentum
balance solver solved using finite element methods (Tezaur et al., 2015). Ice velocity is solved on a two-dimensional map
plane triangulation extruded vertically to form tetrahedra. Mass and tracer transport occur on the Voronoi dual mesh using a
mass-conserving finite volume first-order upwinding scheme. Mesh resolution is 2 km along grounding lines and in all marine
regions of West Antarctica and in marine regions of East Antarctica where present day ice thickness is less than 2500 m to
ensure that the grounding line remains in the fine resolution region even under full retreat of West Antarctica and large parts of
East Antarctica. Mesh resolution coarsens to 20 km in the ice sheet interior and no greater than 6 km in the large ice shelves.
The horizontal mesh has 1.6 million cells. The mesh uses 10 vertical layers that are finest near the bed (4% of total thickness
in deepest layer) and coarsen towards the surface (23% of total thickness in shallowest layer). Ice temperature is based on



results from Van Liefferinge and Pattyn (2013) and held fixed in time. The model uses a linear basal friction law with spatially-varying basal friction coefficient. The basal friction of grounded ice and the viscosity of floating ice are inferred to best match observed surface velocity (Rignot et al., 2011) using an adjoint-based optimization method (Perego et al., 2014) and then kept

constant in time. The grounding line position is determined using hydrostatic equilibrium, with sub-element parameterization of the friction. Sub-ice-shelf melt rates come from Rignot et al. (2013) and are extrapolated across the entire model domain to provide non-zero ice shelf melt rates after grounding line retreat. The surface mass balance is from RACMO2.1 1979-2010 mean (Lenaerts et al., 2012). Maps of surface and basal mass balance forcing are kept constant with time in ctrl_proj experiment. Time-varying anomalies of surface and basal mass balance relative to the original fields are applied in all other

experiments. The ice front position is fixed at the extent of the present-day ice sheet. After initialization, the model is relaxed for 99 years, so that the geometry and grounding lines can adjust.

**ILTS_PIK_SICOPOLIS1**

The model SICOPOLIS version 5.1 (www.sicopolis.net) is applied to the Antarctic ice sheet with hybrid shallow-ice–shelfy-stream dynamics for grounded ice (Bernales et al., 2017) and shallow-shelf dynamics for floating ice. Ice thermodynamics

is treated with the melting-CTS enthalpy method (ENTM) by Greve and Blatter (2016). The ice surface is assumed to be traction-free. Basal sliding under grounded ice is described by a Weertman-Budd-type sliding law with sub-melt sliding (Sato and Greve, 2012) and subglacial hydrology (Kleiner and Humbert, 2014; Calov et al., 2018). The model is initialized by a paleoclimatic spin-up over 140000 years until 1990, forced by Vostok $\delta$D converted to $\Delta T$ (Petit et al., 1999), in which the topography is nudged towards the present-day topography to enforce a good agreement (Rückamp et al., 2018). The basal

sliding coefficient is determined individually for the 18 IMBIE-2016 basins (Rignot and Mouginot, 2016) by minimizing the RMSD between simulated and observed logarithmic surface velocities. The historical run from 1990 until 2015 employs the NorESM1-M-RCP8.5 atmospheric and oceanic forcing. For the last 2000 years of the spin-up, the historical run and the future climate simulations, a regular (structured) grid with 8 km resolution is used. In the vertical, we use terrain-following coordinates with 81 layers in the ice domain and 41 layers in the thermal lithosphere layer below. The present-day surface temperature is

parameterized (Fortuin and Oerlemans, 1990), the present-day precipitation is by Arthern et al. (2006) and Le Brocq et al. (2010), and runoff is modelled by the positive-degree-day method with the parameters by Sato and Greve (2012). The 1960–1989 average SMB correction that results diagnostically from the nudging technique is used as a prescribed SMB correction for the future climate simulations. The bed topography is Bedmap2 (Fretwell et al., 2013), the geothermal heat flux is by Martos et al. (2017), and isostatic adjustment is included using an elastic-lithosphere–relaxing-asthenosphere (ELRA) model

(parameters by Sato and Greve, 2012). Present-day ice-shelf basal melting is parameterized by the ISMIP6 standard approach (Eq. (1)). A more detailed description of the set-up (which is consistent with the one used for the LARMIP-2 (Levermann et al., 2019) and ABUMIP (Sun et al., J. Glaciol., in preparation) initiatives) will be given elsewhere (Greve et al., Geosci. Model Dev., in preparation).





**IMAU_IMAUICE**

The finite difference model (de Boer et al., 2014) uses a combination of SIA and SSA solutions, with velocities added over grounded ice to model basal sliding (Bueler and Brown, 2009). The model grid at 32 km horizontal resolution covers the entire Antarctic ice sheet and surrounding ice shelves. The grounded ice margin is freely evolving, while the shelf extends to the grid margin and a calving front is not explicitly determined. We use the Schoof flux boundary condition (Schoof, 2007) at the grounding line with a heuristic rule following Pollard and DeConto (2012b). For the ISMIP6 projections the sea

level equation is not solved or coupled (de Boer et al., 2014). We run the thermodynamically coupled model with constant present-day boundary conditions to determine a thermodynamic steady state. The model is first initialised for 100 kyr using the average 1979-2014 SMB and surface ice temperature from RACMO 2.3 (van Wessem et al., 2014). Bedrock elevation is fixed in time with data taken from the Bedmap2 dataset (Fretwell et al., 2013), and geothermal heat flux data are from (Shapiro and Ritzwoller, 2004). We then run for 30 kyr with constant ice temperature from the first run to get to a dynamic steady state,

which was our initial condition for initMIP. For IMAUICE1 we assign this steady state to the year 1978 and run the historical period 1979-2014 unforced, keeping the initial SMB constant and sub-shelf basal melting at zero. This model setup is provided for comparison with initMIP. For IMAUICE2 we assign the steady state to the year 1900 and run a 79 year experiment with constant SMB and sub-shelf basal melt rates estimated for the modelled ice draft at 1900 using the shelf melt parameterization of Lazeroms et al. (2018) with a thermal forcing derived from the WOA at 400 m depth. We continue with the historical

period 1979-2014, keeping the initial sub-shelf basal melt rates constant, with transient SMB variations from RACMO 2.3 (van Wessem et al., 2014).

**JPL_ISSM**

The JPL_ISSM ice sheet model configuration relies on data assimilation of present-day conditions, followed by a short model relaxation as described in Schlegel et al. (2018). The model domain covers present-day Antarctic Ice Sheet, and its geometry

is based on an early version of BedMachine Antarctica (Morlighem et al., 2019a). The model is based on the 2D Shelfy-Stream Approximation (MacAyeal, 1989), and the mesh resolution varying between 1 km along the coast to 50 km in the interior, and a resolution of 8 km or finer within the boundary of all initial ice shelves. The model is vertically extruded into 15 layers. To estimate land ice viscosity ($B$), we compute the ice temperature based on a thermal steady state (Seroussi et al., 2013), using a three dimensional higher-order (Blatter, 1995; Pattyn, 2003) stress balance equations, observations of surface

velocities (Rignot et al., 2011), and basal friction inferred from surface elevations (Morlighem et al., 2010). Thermal boundary conditions are geothermal heat flux from Maule et al. (2005) and surface temperatures from Lenaerts et al. (2012). Steady state ice temperatures are then vertically averaged and used to calibrate the ice viscosity, which is held constant over time. To infer the unknown basal friction coefficient over grounded ice and the ice viscosity of the floating ice, we use data assimilation (MacAyeal, 1993; Morlighem et al., 2010), to reproduce observed surface velocities from Rignot et al. (2011). Then, we run

the model forward for 2 years, allow the grounding line position and ice geometry to relax (Seroussi et al., 2011; Gillet-Chaulet et al., 2012). The grounding line evolves assuming hydrostatic equilibrium and following a sub-element grid scheme (SEP2





in Seroussi et al., 2014). The ice front remains fixed in time during all simulations performed, and we impose a minimum ice thickness of 1 m everywhere in the domain. The surface mass balance and the ice shelf basal melt rates used in the control experiment are respectively from the 1979-2010 mean of RACMO2.1 (Lenaerts et al., 2012) and from the 2004-2013 mean
after Schodlok et al. (2016).

**LSCE_GRISLI**

The GRISLI model is a three-dimensional thermo-mechanically coupled ice sheet model originating from the coupling of the inland ice model of Ritz (1992) and Ritz et al. (1997) and the ice shelf model of Rommelaere (1996), extended to the case of ice streams treated as dragging ice shelves (Ritz et al., 2001). In the version used here, over the whole domain, the velocity
field consists in the superposition of the shallow-ice approximation (SIA) velocities for ice flow due to vertical shearing and the shallow-shelf approximation (SSA) velocities, used as a sliding law (Bueler and Brown, 2009). For the initMIP-Antarctica experiments, we used the GRISLI version 2.0 (Quiquet et al., 2018) which includes the analytical formulation of Schoof (2007) to compute the flux at the grounding line. Basal drag is computed with a power-law basal friction (Weertman, 1957). For this study, we use an iterative inversion method to infer a spatially variable basal drag coefficient that insures an ice thickness as
close as possible to observations with a minimal model drift (Le Clec'h et al., 2019). The basal drag is assumed to be constant for the forward experiments.

     The model uses finite differences on a staggered Arakawa C-grid in the horizontal plane at 16 km resolution with 21 vertical levels. Atmospheric forcing, namely near-surface air temperature and surface mass balance, is taken from the 1979-2016 climatological annual mean computed by RACMO2.3p2 regional atmospheric model (van Wessem et al., 2018). Sub-shelf
basal melting rates are computed with the non-local quadratic parametrization suggested in ISMIP. For the inversion step and the control experiments we use the 1995-2017 climatological observed thermal forcing. The initial ice sheet geometry, bedrock and ice thickness, is taken from the Bedmap2 dataset (Fretwell et al., 2013) and the geothermal heat flux is from Shapiro and Ritzwoller (2004).

**NCAR_CISM**

The Community Ice Sheet Model (CISM, Lipscomb et al., 2019) uses finite element methods to solve a depth-integrated higher-order approximation (Goldberg, 2011) over the entire Antarctic ice sheet. The model uses a structured rectangular grid with uniform horizontal resolution of 4 km and 5 vertical $\sigma$−coordinate levels. The ice sheet is initialized with present-day geometry and an idealized temperature profile, then spun up for 30,000 years using 1979-2016 climatological surface mass balance and surface air temperature from RACMO2.3 (van Wessem et al., 2018). During the spin-up, basal friction parameters (for grounded
ice) and sub-shelf melt rates (for floating ice) are adjusted to nudge the ice thickness during present-day observations. This method is a hybrid approach between assimilation and spin-up, similar to that described by Pollard and DeConto (2012a). The geothermal heat flux is taken from Shapiro and Ritzwoller (2004). The basal sliding is similar to that of Schoof (2005), combining power-law and Coulomb behavior. The grounding line location is determined using hydrostatic equilibrium and sub-element parameterization (Gladstone et al., 2010; Leguy et al., 2014). Basal melt is applied in partly floating grid cells



in proportion to the floating fraction as determined by the grounding-line parameterization. The calving front is initialized
from present-day observations and thereafter is allowed to retreat but not advance. For the historical run (1995–2014), the
SMB anomaly was provided by RACMO2.3, and the basal melt rate anomaly was derived from NorESM1-M RCP8.5 thermal
forcing. For the open parameterization of basal melting, we weighted the melt from the standard non-local parameterization
by $\sin \theta$, where $\theta$ is the ice shelf basal slope angle, with $\gamma_0$ recalibrated by N. Jourdain. See Lipscomb et al. (2019) for more
information about the model.

**PIK_PISM**

With the Parallel Ice Sheet Model (PISM, Bueler and Brown, 2009; Winkelmann et al., 2011, www.pism-docs.org, version
1.0), we perfom an equilibrium simulation on a regular rectangular grid with 8 km horizontal resolution. The vertical resolu-
tion increases from 100 m at the top of the domain to 13 m at the (ice) base, with a domain height of 6000 m. PISM uses a
hybrid of the Shallow-Ice Approximation (SIA) and the two-dimensional Shelfy-Stream Approximation of the stress balance
(SSA, MacAyeal, 1989; Bueler and Brown, 2009) over the entire Antarctic Ice Sheet. The grounding line position is deter-
mined using hydrostatic equilibrium, with sub-grid interpolation of the friction at the grounding line (Feldmann et al., 2014).
The calving front position can freely evolve using the Eigencalving parameterization (Levermann et al., 2012). PISM is a
thermomechanically-coupled (polythermal) model based on the Glen-Paterson-Budd-Lliboutry-Duval flow law (Aschwanden
et al., 2012). The three-dimensional enthalpy field can evolve freely for given boundary conditions.

The model is initialized from Bedmap2 geometry (Fretwell et al., 2013), with surface mass balance and surface temperatures
from RACMOv2.3 1986-2005 mean (van Wessem et al., 2014) remapped from 27 km resolution. Geothermal heat flux is from
Shapiro and Ritzwoller (2004). We use the Potsdam Ice-shelf Cavity model (PICO, Reese et al., 2018a) which extends the
ocean box model by Olbers and Hellmer (2010) for application in three dimensional ice-sheet models to calculate basal melt
rate patterns underneath the ice shelves. We use a compilation of observed ocean temperature and salinity values (1979-
2013, Schmidtko et al., 2014) (1955-2010, Locarnini et al., 2019) to drive PICO. We apply a power law for sliding with a
Mohr–Coulomb criterion relating the yield stress to parameterized till material properties and the effective pressure of the
overlaying ice on the saturated till (Bueler and van Pelt, 2015).Basal friction and sub-shelf melting are linearly interpolated
on a sub-grid scale around the grounding line (Feldmann et al., 2014). We apply eigen-calving (Levermann et al., 2012) in
combination with the removal of all ice that is thinner than $50 \, \mathrm{m}$ or extends beyond present-day ice fronts (Fretwell et al.,
2013).

**UCIJPL_ISSM**

We initialize the model by using data assimilation of present day conditions, following the method presented in Morlighem
et al. (2013). The mesh horizontal resolution varies from 3 km near the margins to 30 km inland where the ice is almost
stagnant. The mesh is vertically extruded into 10 layers. We use a Higher-Order stress balance (Pattyn, 2003) and an Enthalpy
based thermal model (Aschwanden et al., 2012; Seroussi et al., 2013). The initialization is a two-step process: we first invert
for ice shelf viscosity ($B$), and then invert for basal friction under grouded ice assuming thermo-mechanical steady state.



Our geometry is based on BedMachine Antarctica (Morlighem et al., 2019a). The thermal model is constrained by surface temperatures from Comiso (2000) and geothermal heat flux from Shapiro and Ritzwoller (2004), both included in the SeaRISE dataset (Shapiro and Ritzwoller, 2004; Nowicki et al., 2013a). The surface mass balance used in the control experiment is from RACMO 2.3 (van Wessem et al., 2014).

**ULB_FETISH**

The f.ETISh (fast Elementary Thermomechanical Ice Sheet) model (Pattyn, 2017) version 1.3 is a vertically integrated hybrid finite-difference (SSA for basal sliding; SIA for grounded ice deformation) ice sheet/ice shelf model with vertically-integrated thermomechanical coupling. The transient englacial temperature field is calculated in a 3d fashion. The marine boundary is represented by a grounding-line flux condition according to (Schoof, 2007), coherent a power-law basal sliding (power-law coefficient of 2). Model initialization is based on an adapted iterative procedure based on Pollard and DeConto (2012a) to fit the model as close as possible to present-day observed thickness and flow field (Pattyn, 2017). The model is forced by present-day surface mass balance and temperature (van Wessem et al., 2014), based on the output of the regional atmospheric climate model RACMO2 for the period 1979-2011. The PICO model (Reese et al., 2018a) was employed to calculate sub-shelf melt rates, based on present-day observed ocean temperature and salinity (Schmidtko et al., 2014) on which the initMIP forcings for the different basins are added. The model is run on a regular grid of 16 km with time steps of 0.05 year.

**UTAS_ElmerIce**

The Elmer/Ice model domain covers the present-day Antarctic Ice Sheet, and its geometry is interpolated from the Bedmap2 dataset (Fretwell et al., 2013). An unstructured mesh in the horizontal is refined using the Hessian of the observed surface velocity, as in Zhao et al. (2018). Mesh resolution in the horizontal varies from approximately 4 km near the grounding lines of fast flowing ice streams to approximately 40 km in the interior. The mesh is extruded to 10 layers in the vertical. The forward simulations solve the Stokes equations directly (Gagliardini et al., 2013). Initialisation comprised the following steps:

1. Short surface relaxation (20 timesteps of 0.001 years).

2. Inversion for sliding coefficient with constant temperature $T = -20\,\mathrm{C}$ (Gillet-Chaulet et al., 2016).

3. Steady state temperature simulation using the flow field from previous step.

4. Inversion for sliding coefficient using the new temperature field from the previous step.

5. Thermo-mechanically coupled steady state temperature-velocity calculation using the basal sliding coefficient distribution from the previous step.

6. Inversion for sliding coefficient using the latest temperature field from the previous step.

7. Surface relaxation (10 years with an increasing timestep size).

A linear sliding relation is used. The ice front is not allowed to evolve. Elmer/Ice solves a contact problem at the grounding line, and no further parameterisations are applied. Thermal boundary conditions are geothermal heat flux from Maule et al.



(2005) and surface temperatures from Comiso (2000). Steady temperature is solved for during the initialisation steps and held

constant during the transient simulations. We impose a minimum ice thickness of 40 m everywhere in the domain. The surface mass balance used in the surface relaxation and control experiment is the 1995 to 2014 mean from the MAR model (Agosta et al., 2019). Basal melt rates are computed using the local quadratic parameterisation provided by ISMIP as an alternative to the non-local parameterisation.

### VUW_PISM

We use an identical approach to the one described in Golledge et al. (2019). Starting from initial bedrock and ice thickness conditions from Morlighem et al. (2019a), together with reference climatology from van Wessem et al. (2014) we run a multi-stage spinup that guarantees well-evolved thermal and dynamic conditions without loss of accuracy in terms of geometry. This is achieved through an iterative nudging procedure, in which incremental grid refinement steps are employed that also include resetting of ice thicknesses to initial values. Drift is thereby eliminated, but thermal evolution is preserved by remapping of

temperature fields at each stage. In summary, we start with an initial 32 km resolution 20 year smoothing run in which only the shallow-ice approximation is used. Then, holding the ice geometry fixed, we run a 250000 year, 32 km resolution, thermal evolution simulation in which temperatures are allowed to equilibrate. Refining the grid to 16 km and resetting bed elevations and ice thicknesses we run a further 1000 years using full model physics and a present-day climate, then refine the grid to 10 km for a further 500 years, then refine the grid to 8 km for a GCM-forced historical run from 1950 to 2000. The resultant

configuration is then used as the starting point for each of our forward experiments.

### VUB_AISMPALEO

The Antarctic ice sheet model from the Vrije Universiteit Brussel is derived from the coarse-resolution version used mainly in simulations of the glacial cycles (Huybrechts, 1990, 2002). It considers thermomechanically coupled flow in both the ice sheet and the ice shelf, using the SIA/SSA coupled across a transition zone one grid cell wide. Basal sliding is calculated using a

Weertman relation inversely proportional to the height above buoyancy wherever the ice is at the pressure melting point. The horizontal resolution is 20 km, and there are 31 layers in the vertical. The model is initialized with a freely evolving geometry until a steady state is reached. The precipitation pattern is based on the Giovinetto and Zwally (2000) compilation used in Huybrechts et al. (2000), updated with accumulation rates obtained from shallow ice cores during the EPICA pre-site surveys (Huybrechts, 2007). Surface melting is calculated over the entire model domain with the PDD scheme, including meltwater

retention by refreezing and capillary forces in the snowpack (Janssens and Huybrechts, 2000). The sub-shelf basal melt rate is parameterized as a function of local mid-depth (485-700 m) ocean-water temperature above the freezing point (Beckmann and Goosse, 2003). A distinction is made between protected ice shelves (Ross and Filchner-Ronne) with a low melt factor and all other ice shelves with a higher melt factor. Ocean temperatures are derived from the LOVECLIM climate model (Goelzer et al., 2016), and parameters are chosen to reproduce observed average melt rates (Depoorter et al., 2013). Heat conduction is

calculated in a slab of bedrock 4 km thick underneath the ice sheet. Isostatic compensation is based on an elastic lithosphere





floating on a viscous asthenosphere (ELRA model) but is not allowed to evolve further in line with the initMIP-Antarctica experiments

*Competing interests.* Eric Larour serves as topical editor for the Journal. William Lipscomb, Sophie Nowicki, Helene Seroussi, Ayako Abe-Ouchi, and Robin Smith are editors of the special issue The Ice Sheet Model Intercomparison Project for CMIP6 (ISMIP6).

*Acknowledgements.* We thank the Climate and Cryosphere (CliC) effort, which provided support for ISMIP6 through 5 sponsoring of workshops, hosting the ISMIP6 website and wiki, and promoted ISMIP6. We acknowledge the World Climate Research Programme, which, through its Working Group on Coupled Modelling, coordinated and promoted CMIP5 and CMIP6. We thank the climate modeling groups for producing and making available their model output, the Earth System Grid Federation (ESGF) for archiving the CMIP data and providing access, the University at Buffalo for ISMIP6 data distribution and upload, and the multiple funding agencies who support CMIP5 and CMIP6 and ESGF. We thank the ISMIP6 steering committee, the ISMIP6 model selection group and ISMIP6 dataset preparation group for their continuous engagement in defining ISMIP6. This is ISMIP6 contribution No X.

Research was carried out at the Jet Propulsion Laboratory, California Institute of Technology. Helene Seroussi and Nicole Schlegel are supported by grants from NASA Cryospheric Science and Modeling, Analysis, Predictions Programs. AB was supported by the U.S. Department of Energy (DOE) Office of Science Regional and Global Model Analysis (RGMA) component of the Earth and Environmental System Modeling (EESM) program (HiLAT-RASM project), and the DOE Office of Science (Biological and Environmental Research), Early Career Research program. Heiko Goelzer has received funding from the programme of the Netherlands Earth System Science Centre (NESSC), financially supported by the Dutch Ministry of Education, Culture and Science (OCW) under grant no. 024.002.001. Rupert Gladstone and Thomas Zwinger were supported by Academy of Finland grants 286587 and 322430. Chen Zhao was supported under Australian Research Council's Special Research Initiative for Antarctic Gateway Partnership (Project ID SR140300001). Support for Xylar Asay-Davis, Matthew Hoffman, Stephen Price, and Tong Zhang was provided through the Scientific Discovery through Advanced Computing (SciDAC) program funded by the US Department of Energy (DOE), Office of Science, Advanced Scientific Computing Research and Biological and Environmental Research Programs. MALI simulations used resources of the National Energy Research Scientific Computing Center, a DOE Office of Science user facility supported by the Office of Science of the U.S. Department of Energy under Contract DE-AC02-05CH11231. Nicolas Jourdain is funded by the French National Research Agency (ANR) through the TROIS-AS project (ANR-15-CE01-0005-01) and by the European Commission through the TiPACCs project (grant 820575, call H2020-LC-CLA-2018-2). Philippe Huybrechts and Jonas Van Breedam acknowledge support from the iceMOD project funded by the Research Foundation - Flanders (FWO-Vlaanderen). Ralf Greve was supported by the Japan Society for the Promotion of Science (JSPS) KAKENHI grant numbers JP16H02224, JP17H06104 and JP17H06323.



Support for Nicholas Golledge and Daniel Lowry was provided by the New Zealand Ministry of Business Innovation and Employment contract RTVU1705. The work of Thomas Kleiner has been conducted in the framework of the PalMod project (FKZ: 01LP1511B), supported by the German Federal Ministry of Education and Research (BMBF) as Research for Sustainability initiative (FONA). Support for Mathieu Morlighem and Tyler Pelle was provided by the National Science Foundation (NSF: Grant 1739031). Development of PISM is supported by NASA grant NNX17AG65G and NSF grants PLR-1603799 and PLR-1644277. The authors gratefully acknowledge the European Re-

gional Development Fund (ERDF), the German Federal Ministry of Education and Research and the Land Brandenburg for supporting this project by providing resources on the high performance computer system at the Potsdam Institute for Climate Impact Research. Computer resources for this project have been also provided by the Gauss Centre for Supercomputing/Leibniz Supercomputing Centre (www.lrz.de) under Project-ID pr94ga and pn69ru. R.R. was supported by the Deutsche Forschungsgemeinschaft (DFG) by grant WI 4556/3-1. T.A. is supported by the Deutsche Forschungsgemeinschaft (DFG) in the framework of the priority program "Antarctic Research with comparative

investigations in Arctic ice areas" by grant WI4556/4-1. Reinhard Calov was funded by the Bundesministerium für Bildung und Forschung (BMBF) grants PalMod-1.1 and PalMod-1.3. Gunter Leguy and William Lipscomb were supported by the National Center for Atmospheric Research, which is a major facility sponsored by the National Science Foundation under Cooperative Agreement No. 1852977. Computing and data storage resources for CISM simulations, including the Cheyenne supercomputer (doi:10.5065/D6RX99HX), were provided by the Computational and Information Systems Laboratory (CISL) at NCAR.



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

©c Author(s) 2020. CC BY 4.0 License.





**Figure 1.** Evolution of surface mass balance (a, in Gt/yr), basal melt rate (b, in Gt/yr), and volume above floatation (c, in Gt) during the historical and ctrl_proj experiments for all the simulations performed with the open and standard framework.



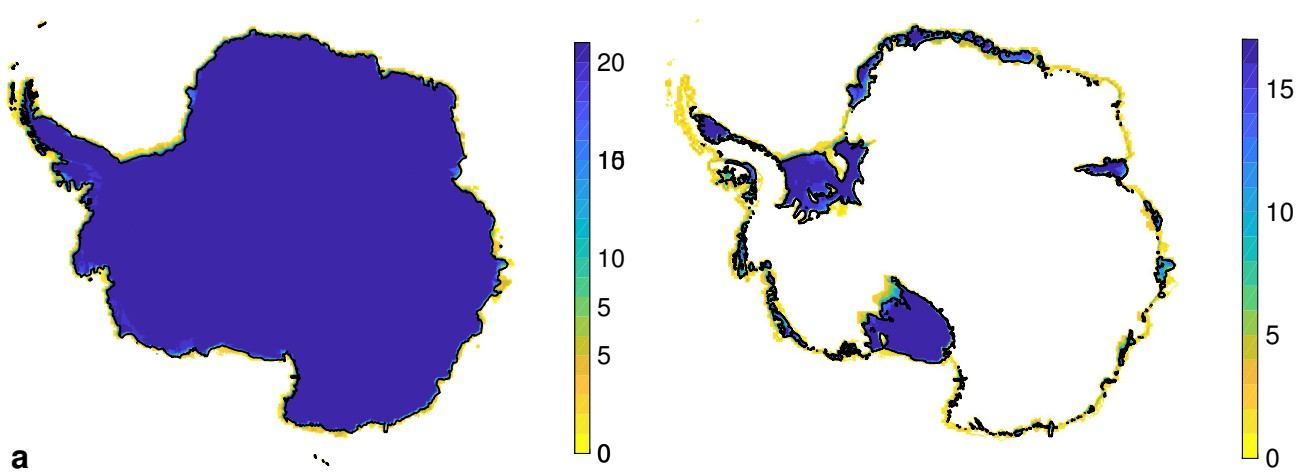

**Figure 2.** Total (left) and floating (right) ice extent at the beginning of the experiments (January 2015). Colors indicate the number of models simulating total ice (left) and floating ice (right) extent at every point of the 8-km grid. Black lines are observations of the total and floating ice extent, respectively (Morlighem et al., 2019a).

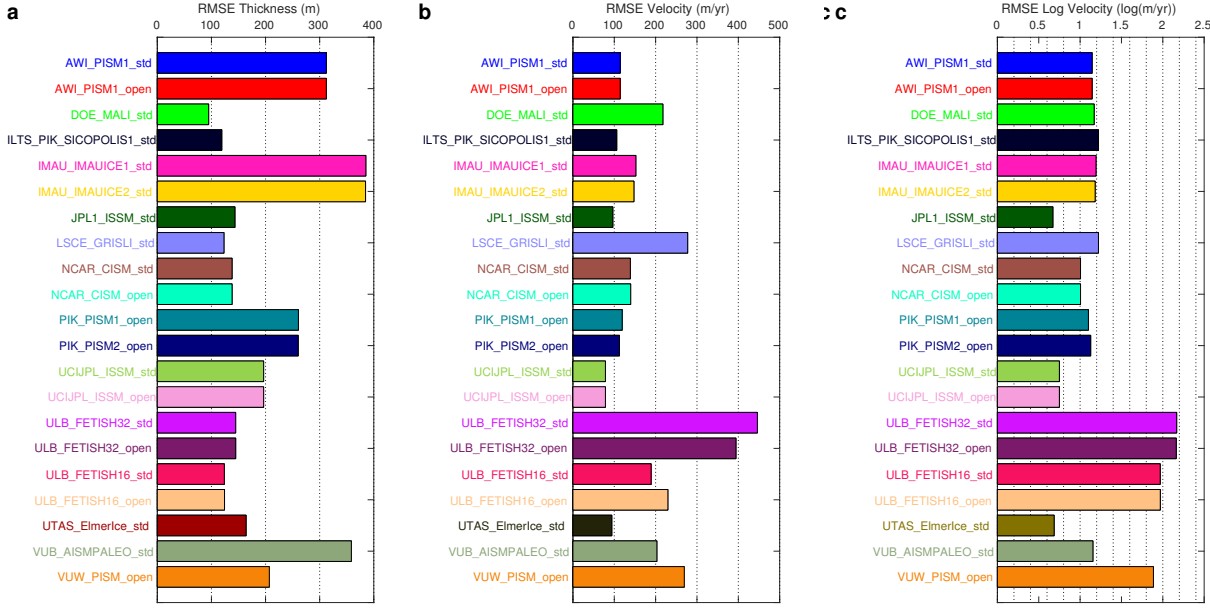

**Figure 3.** Root Mean Square Error in ice thickness (a, in m), ice velocity (b, in m/yr), and logarithm of ice velocity (c, in log(m/yr)) between modeled and observed values at the beginning of the experiments (January 2015).



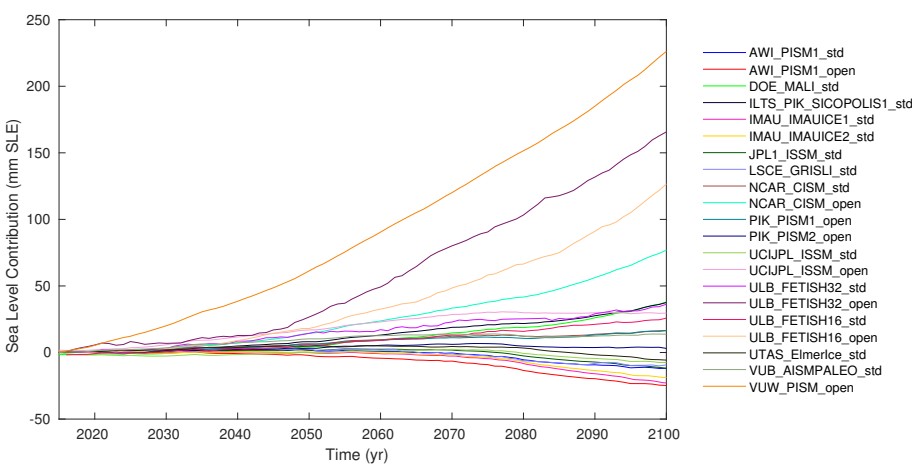

**Figure 4.** Evolution of ice volume above floatation (in mm SLE) over 2015–2100 from NorESM1-M RCP 8.5 scenario (exp01 and exp05) relative to ctrl_proj.

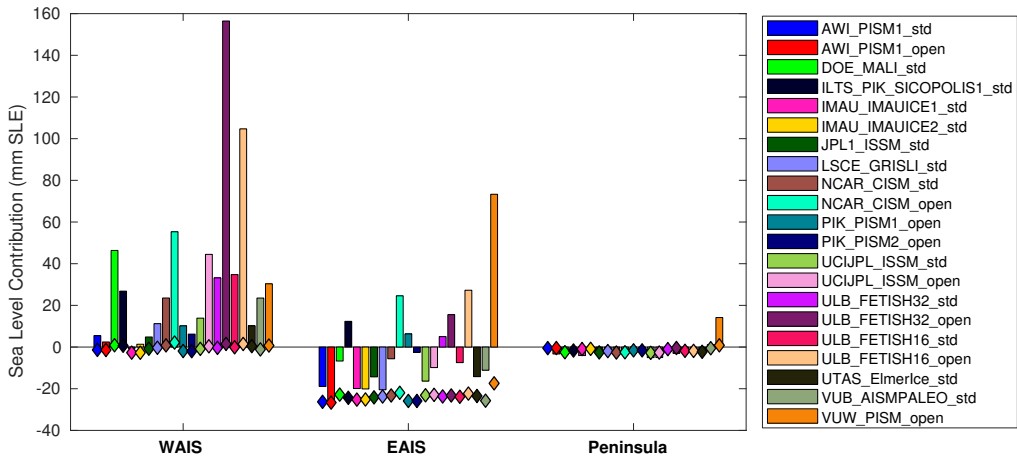

**Figure 5.** Regional change in volume above floatation (in mm SLE) and integrated SMB changes (diamond shapes, in mm SLE) for the 2015-2100 period under medium forcing from NorESM1-M RCP 8.5 scenario (exp01 and exp05) relative to ctrl_proj.





**Figure 6.** Mean (a and b) and standard deviation (c and d) of simulated thickness change (a and c, in m) and velocity change (b and d, in m/yr) between 2015 and 2100 under medium forcing from NorESM1-M RCP 8.5 scenario (exp01 and exp05) relative to ctrl_proj. .





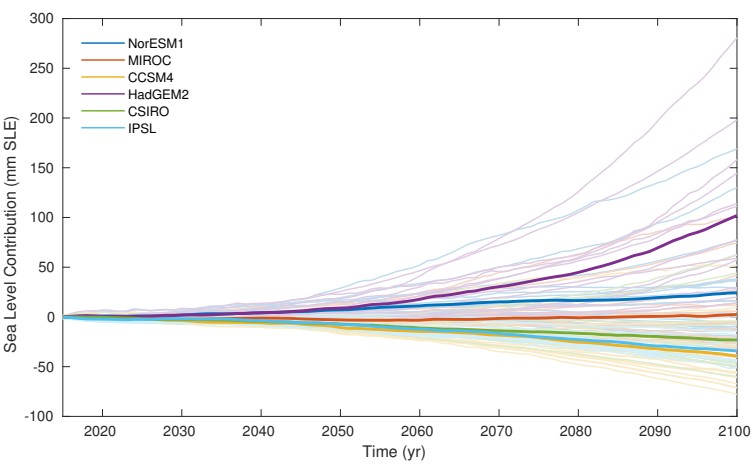

**Figure 7.** Evolution of ice volume above floatation (in mm SLE) over 2015–2100 period with medium forcing from the six CMIP5 AOGCMs and RCP 8.5 scenario relative to ctrl_proj. Thin lines show results from individual ice sheet model simulations, and thick lines show mean values averaged for each AOGCM.

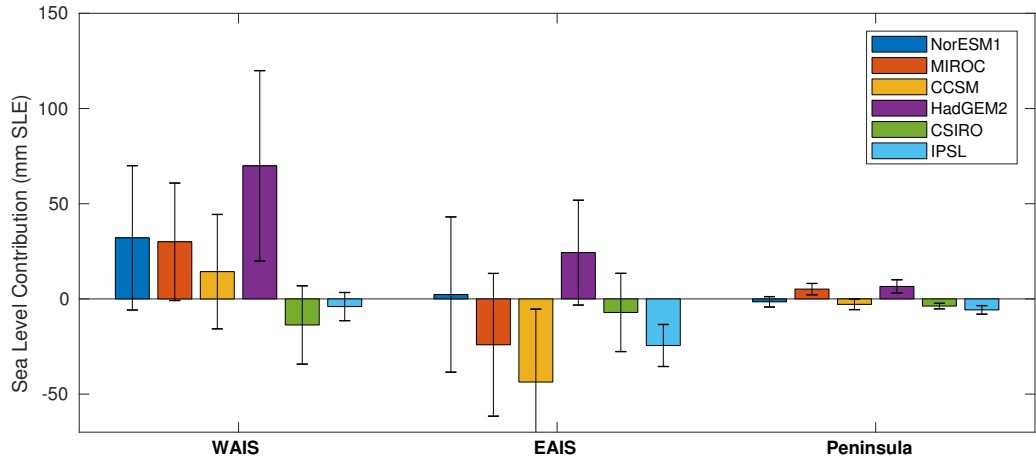

**Figure 8.** Regional change in volume above floatation (in mm SLE) for 2015–2100 from six CMIP5 AOGCMs under the RCP 8.5 scenario with median forcing, relative to ctrl_proj. Black lines show the standard deviation.





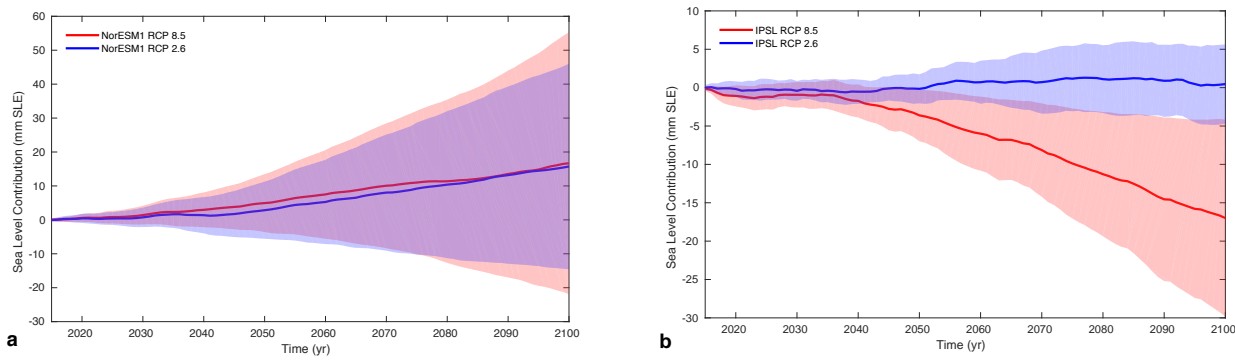

**Figure 9.** Impact of RCP scenario on projected evolution of ice volume above floatation for the NorESM1-M (a) and IPSL (b) AOGCMs. Red and blue curves show mean evolution for RCP 8.5 and RCP 2.6, respectively, and shaded background the standard deviation.







**Figure 10.** Regional change in volume above floatation (in mm SLE) and integrated SMB changes (diamond shapes, in mm SLE) for 2015–2100 under RCP 8.5 (red) and RCP 2.6 (blue) scenario forcing from NorESM1-M (a) and IPSL (b) relative to ctrl_proj from individual model simulations.





**Figure 11.** Regional change in integrated basal melt (a, in Gt) and volume above floatation (b, in mm SLE) for 2015–2100 under medium forcing from the six CMIP5 AOGCMs using RCP 8.5 forcing, relative to ctrl_proj for the open and standard basal melt frameworks. Black lines show the standard deviations.


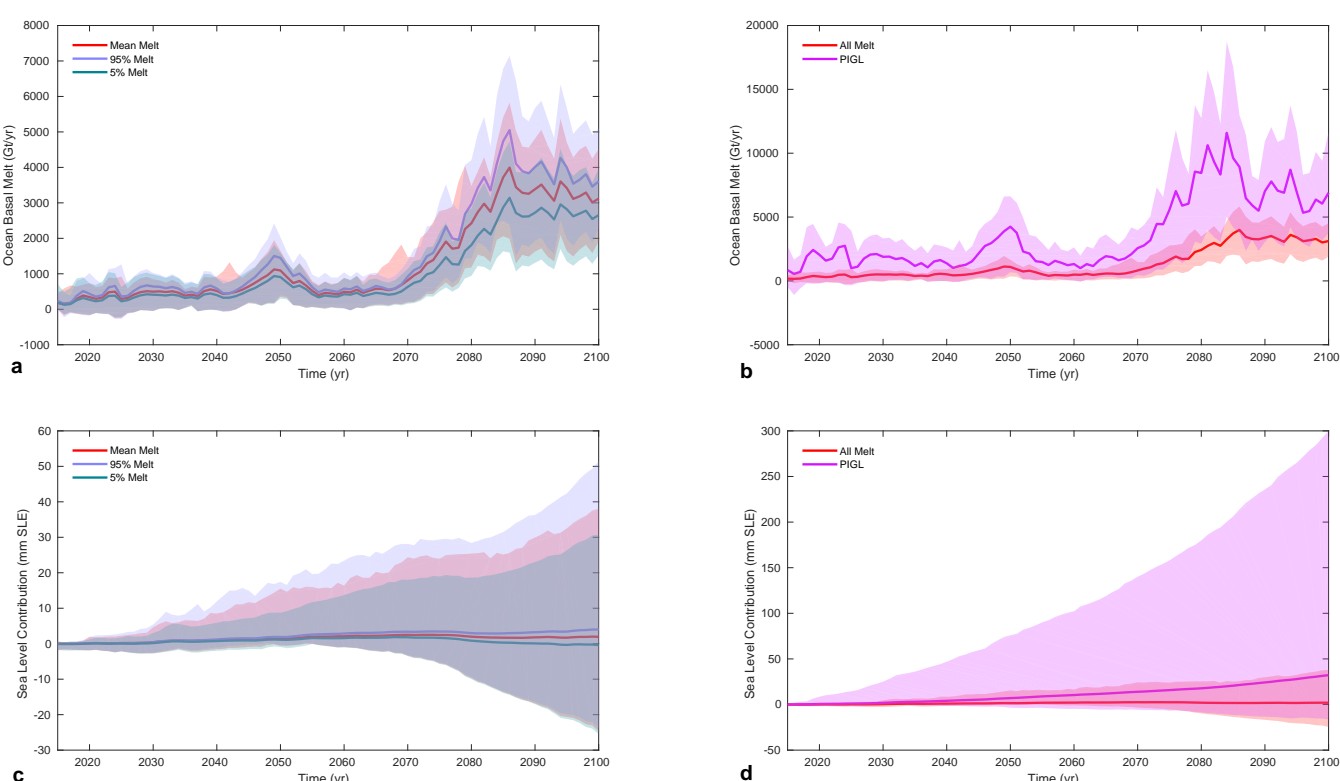

**Figure 12.** Impact of basal melt parameterization (a and c, $5^{th}$-, $50^{th}$- and $95^{th}$- percentile values of $\gamma_0$ distribution) and calibration (b and d, "MeanAnt" and "PIGL" calibrations) on basal melt evolution (a and b, in Gt/yr) and ice volume above floatation relative to ctrl_proj (c and d, in mm SLE) over 2015–2100. Lines show the mean values and shaded background the simulations spread. Note that the y-axis differs in all plots.



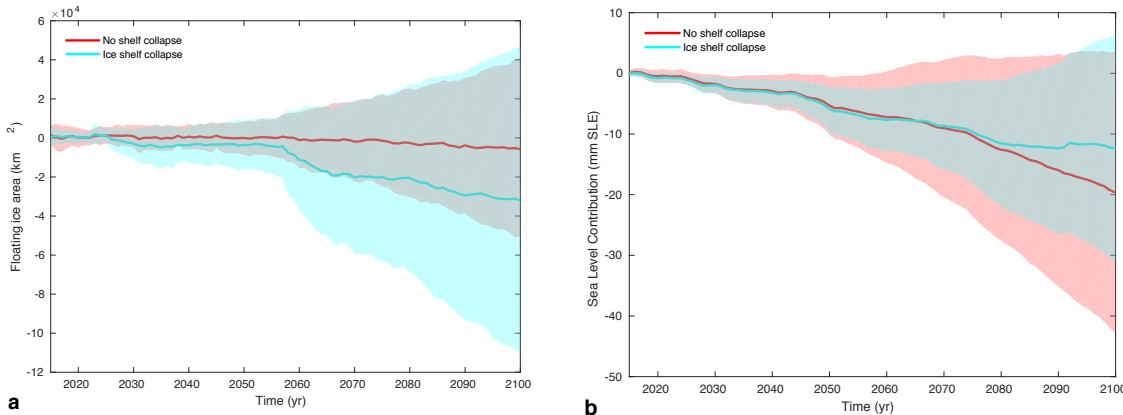

**Figure 13.** Evolution of basal melt (a, in Gt/yr) and ice volume above floatation relative to ctrl_proj (b, in mm SLE) without (red) and with (cyan) ice shelf collapse over the 2015-2100 period under the CCSM4 RCP 8.5 forcing. Lines show the mean values and shaded background the standard deviations.

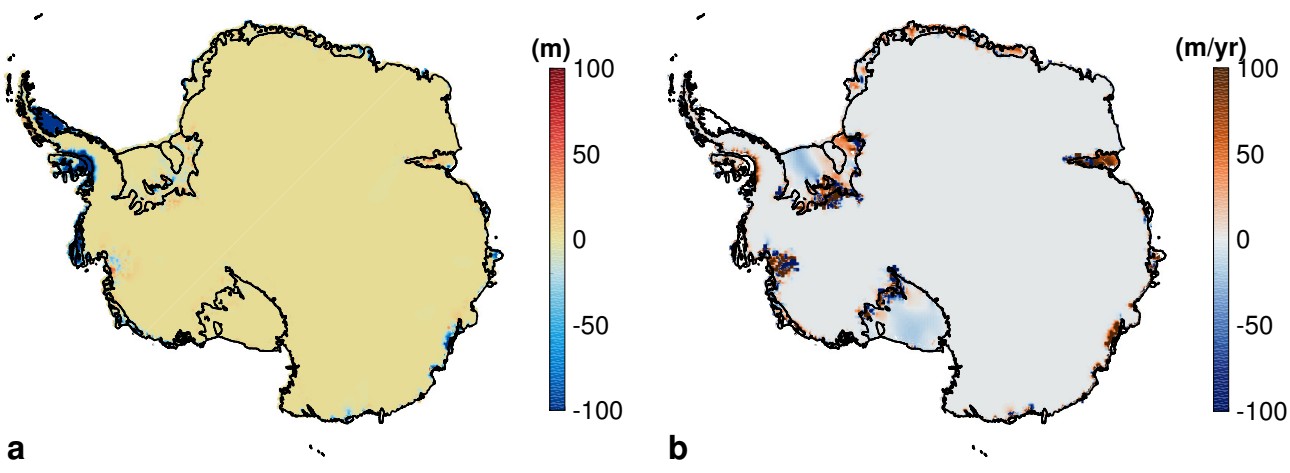

**Figure 14.** Mean simulated thickness change (a, in m) and velocity change (b, in m/yr) between 2015 and 2100 with ice shelf collapse under CCSM4 RCP 8.5 scenario (exp11 and exp12) relative to similar experiments without ice shelf collapse (exp04 and exp08).



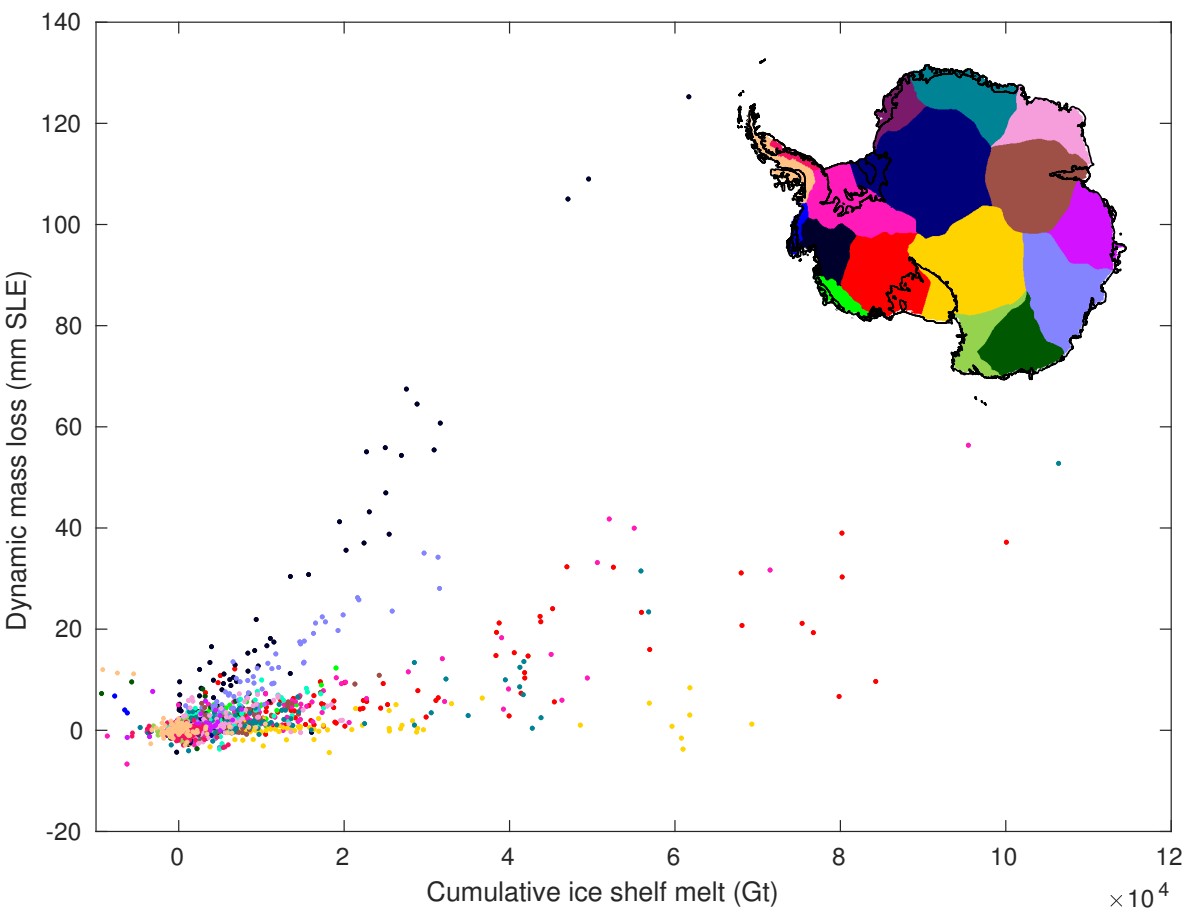

**Figure 15.** Dynamic mass loss over the 2015-2100 period as a function of cumulative ocean induced basal melt vero the same period for the 18 main Antarctic basins (Rignot et al., 2019) for all RCP 8.5 experiments. Antarctic map shows the location of the 18 basins.