# Peer review of "ISMIP6 Antarctica: a multi-model ensemble of the Antarctic ice sheet evolution over the 21st century"

_The Cryosphere, 2019_

## Referee Comment (RC1) · Anonymous Referee #1 · 6 Mar 2020

The manuscript describes the Ice-Sheet Models Inter-comparison Project for Antarctica. In addition to presenting results, the manuscript also documents various aspects of the project itself. Undoubtedly, it will be published, at some point. The current version, however, requires modifications, restructuring and potential additional analysis (I will come to this later).

The most general comment is that it is not entirely clear who is the intended audience for this manuscript. If it is aimed at wider, more general audience, it is full of jargon and unstated assumptions (e.g. that the ocean temperatures simulated by climate models can be used as a proxy for the sub-ice-shelf melting). If it is primarily aimed at icesheet modellers, it is a little bit thin on results. It would be beneficial for the manuscript if the authors write it with a specific audience in mind. Regardless of that, the text has to be much more clear that the described results are results of simulations, and are not expected contributions of Antarctica to sea level. This seems like an obvious, and redundant comment, however, considering a high profile of this manuscript (a most like reference for the next IPCC report), its language and wording has to be precise. I would recommend to modify all statements similar to "The contribution of the Antarctic ice sheet..." (p.2 line 6), "East Antarctica mass change..." (p.2 line 9), etc. to "The *projected* contribution of the Antarctic ice sheet...", "*Simulated* East Antarctica mass change..." (I'm not going to mark all such phrases, but please correct them all).

Another general comment, which is easy to address, is that it'd be better to use CMs (climate models) instead of AOGCMs. "CM" in CMIP5 stands for "Climate Model", additionally later in the text (lines 85-95) "CM" and "ESM" in names of the models, which outputs were used, indicate the type (complexity) of the model - either a Climate Model or Earth System Model.

Overall, the text has too much jargon (e.g. SMB; it's not clear what "idealized surface mass balance" means). The titles of sections and subsections are too cryptic (e.g. "ctrl_proj", "NorESM1-M RCP 8.5 scenario"). They need to be informative enough to give a general idea of section or subsection content. The subsection "2.1.4 Ice shelf collapse" is misleading in both its title and justification of the experiment. Perhaps it should have quotation marks to indicate the name of an experiment. Lines (157-159), state that hydrofracturing is the main mechanism that leads to an ice-shelf collapse. Though collapse of the Larsen B ice shelf preceded by surface melting, and hydrofracturing was specifically proposed to explain its collapse, it is not the only ice shelf to collapse, and collapses of other ice shelves, for instance Willkins Ice Shelf, were most likely unrelated to hydrofracturing or surface melting (it happened during austral winter). Because hydrofracturing is essentially the only mechanism that can be parameterized in an ice-shelf model, it does not mean that it is the only possible mechanism to trigger

an ice-shelf disintegration. This part of the text needs clarifications.

As mentioned above, the manuscript documents experimental protocols and describes projected Antarctic contributions to sea level. It is unclear whether there will be in-depth analyses of this MIP. Considering the diversity of participated models, it would be interesting to know whether some useful lessons (apart from projected sea-level magnitudes and their spreads) could be learned from this exercise. For instance, because the initial ice sheet geometry affects ice flow and and ice discharge through the grounding line at the later times, it would be interesting to know whether ice-sheet models that use long spin-up as initialization, simulate significantly different ice discharge compared to ice-sheet models used present-day configurations as initial conditions. The same question applies for parameterizations of basal sliding - do models that use inversions of the present-day observations produce different results compared to models that don't employ inverse methods? These are rather suggestions, and it up to the authors to decide what is the scope of the manuscript.

Similarly, the structure of the manuscript is the authors' decision. The current version has a fairly lengthy description of sub-ice-shelf melting simulation. It is not entirely clear why this process has such a prominence compared to other, no better constrained processes (e.g. basal sliding, calving, etc.). As a suggestion, the authors might consider moving details to an appendix, and the current appendix (C at least) to supplemental online materials. The manuscript will benefit from streamlining. Currently, section 2 combines together various unrelated aspects (i.e. kinds of forcing, experiments, etc.). Having a better structure will improve the manuscript readability.

Figures could be more illustrative. Overall, bar figures are difficult to read, simply displaying them in the models' alphabetical order is not informative. The authors might consider modifying the time axis in Fig. 1 (e.g. having uneven spacing prior to 1990s or so) to focus more on a period of time that have results from more models. Panels (b) and (c) in Fig. 3 seem to show the same field but in different units, the purpose of that isn't clear. Perhaps using log scale (for negative values it could a different colormap of

the absolute vales) in figs 6 and 14 might show better spatial variations on the grounded ice sheet, as the largest magnitudes are on ice shelves or in their immediate vicinity. It is unclear how to read fig. 15, its caption does not help with that.

---

## Referee Comment (RC2) · Anonymous Referee #2 · 11 Mar 2020

I commend all authors and contributors for their efforts and time investment into this MIP (one of many) and highly recommend this community effort for publication in TC. I have no significant points of concern; my only main comment is about the discussion and conclusions. Despite the tricky task of analyzing outputs from such a diverse range of forcings and model designs, I would have liked to see a stronger emphasis on the 'lessons learnt' from this exercise, and suggestions for possible ways forward. In my opinion, one of the key messages that should prevail from this MIP is that, despite the large spread in projections, significant advances have been made in the recent decade to reduce the uncertainties. Although this is mentioned in the text, I think this success should be stressed more and perhaps even quantified (e.g. L501 and following, L538-

539). Moreover, as this is very much 'work in progress' whilst the modelling community continues its efforts to improve models, these MIPs are a great way to guide such improvements. Individual groups will have used ISMIP6 and related MIPs to test and upgrade their models, and other developers/users might benefit from adopting these improvements in their own models. Perhaps there is scope for a paragraph or two in the discussion on i) recent key challenges (numerics, physics,...) that have been considered/overcome by individual contributors and how this has influenced their results, ii) an expert judgement on key improvements that need to be prioritized in the near future? In light of future publications, such as additional results based on CMIP6, the community might also want to think about more concrete 'measures of progress'.

Below is a list of more specific comments and points for further clarification.

L1 It might be worth introducing an abbreviation for Antarctic Ice Sheet, as I counted 10 instances on the first 3 pages alone.

L3 'estimated' → estimates of?

L3 You say 'primarily because of differences in the representation of physical processes and the forcings employed' but my understanding is that the initial state of the ice sheet and numerical design of the models are equally important sources of uncertainty?

L4 13 groups?

L7 '...between -7.8 and 30.0cm...': is this for a fixed forcing scenario, or does it include uncertainties from variability in models and the full range of forcings?

L15 define AOGCM

L15 'overall' → additional

L27 'paradigm shift' is rather vague. Perhaps you can be more precise, e.g. by saying that models have been verified against analytical solutions of ice flow, grounding line flow etc.

L30 Do you mean that model validation against observations of past changes is critical to improve projections, or are you alluding to a more general understanding of how climate change affects sea level?

L33-35 Perhaps the ice sheet initial state (and results from initMIP) should be included here as an additional source of uncertainty.

L41 'mitigate the gaps' seems like an unfortunate choice of words. My understanding is that MIPs aim to quantify the spread in model projections, rather than to eliminate the spread?

L44-48 I was expecting to read about the impact of the initial ice sheet state here, but instead the focus is on SMB. Can you provide a 1-line summary of the initMIP results?

L135 Perhaps you can point out that this result was obtained in the context of the idealized MISOMIP experiment, but has not been tested for realistic geometries.

L140-145 Although Jourdain et al. (under review) will provide further details, it would be nice to have a little bit more information here. E.g. it is not clear what is meant by 'random samplings of Antarctic melt rate and ocean temperature'. Are these melt rates from Rignot et al., and is the ocean temperature taken from observations/reanalysis?

L185-186 I'm unclear about the difference between ctrl and ctrl_proj. Are they identical except for the duration, i.e. ctrl runs from the initialization time until 2100, whereas ctrl_proj runs from 2015 until 2100?

Table1, first row. Is 'Ocean coefficient' the \gamma_0 parameter in Eq(1) and what is meant by 'Low', 'Medium' and 'High'?

L201 It would be good to have some further info about the 'open experiments' here. Does it mean that some ocean conditions (T,S,...) are prescribed but the melt parameterization is left free?

L217 include abbreviations FE and FV to help the reader interpret the second column

of Table 3.

L278 what is meant by 'consistent' here? Perhaps this can be quantified.

L278 'ice shelves that extend slightly farther': again, perhaps this can be quantified. Could this be a resolution issue, i.e. the offset is on the order of the resolution of the analysis mesh?

L283-285 this information seems to be repeated from lines 268 and 270.

L300 'trends cannot be considered as a physical response of the AIS...': despite the constant climate conditions applied, could internal ice dynamics not give rise to a trend?

L309 The reference to figure 1 is appropriate here, but I'm finding it hard to distinguish the individual model results due to the choice of colour scheme. It is therefore difficult to verify this point.

L310 please check this sentence as I'm not sure what is meant here.

L312 At this point it is unclear why NorESM1-M was singled out for these experiments. Can you comment?

L325 'slit'?

L332-333, L337-338 The Siple Coast ice streams seem to produce an equally large response, yet they are not mentioned here?

L351 4 out of 6?

L352 You say that 'uncertainties for the WAIS are larger than for the EAIS' but I can't see any significant difference between the length of the error bars in Figure 8...

L376 Both here and later on, it would be useful to reference back to Table 3 with the experiment names, e.g. '...experiments were simulated with both open (exp01-04) and standard (exp05-08)...'

[Figure]

L391 Again, a reference to the experiment names in Table 3 would be helpful, i.e. exp05.

L413 superscript st in 21st

L479 Are ocean processes even reliably included in the Greenland studies?

L499-500 As discussed earlier, it would be nice to see a more quantitative statement here, and further documentation on what is meant by 'significant improvement'.

L538 In my opinion this sentence is somewhat misleading. I assume that the 'main sources of uncertainties' refer to the uncertainties that were addressed as part of this study? Other sources of uncertainty such as the initial state of the ice shelf were not addressed here, and could be equally important.

Table A1. title: FL instead of FX?

Table B1. title: add (2015) to better specify 'beginning of the experiments'?

Figure 2. Something gone wrong with the colorbar in panel a? Also, black lines are very hard to see with the dark blue background, so consider adjusting the colors for more contrast.

Figure 3. Yellow text is hard to read. I'm not sure what the log plot in panel c contributes to the analysis. Spatial maps of ice thickness and velocity std between models might be a useful metric to identify areas where models disagree the most and highlight geographical regions where efforts for improvement should be focused.

Figure 12 and 13. Experiment names in the legend would be a handy cross-reference to Table3 here.

Figure 13b Why is the sea level contribution larger (more negative) without ice shelf collapse?

Figure 15. It is very hard to distinguish individual basins here, whereas this is crucial

to understand the figure. Perhaps consider splitting into subfigures with equal axis to show results for different basins or groups of basins? Also consider adding basin names to help readers understand the main text (L473-475).

---

## Author Comment (AC1) · 17 Apr 2020

**1 Reviewer #1 (Anomymous)**

The manuscript describes the Ice-Sheet Models Inter-comparison Project for Antarctica. In addition to presenting results, the manuscript also documents various aspects of the project itself. Undoubtedly, it will be published, at some point. The current version, however, requires modifications, restructuring and potential additional analysis (I will come to this later).

The most general comment is that it is not entirely clear who is the intended audience for this manuscript. If it is aimed at wider, more general audience, it is full of jargon and unstated assumptions (e.g. that the ocean temperatures simulated by climate models can be used as a proxy for the sub-ice-shelf melting). If it is primarily aimed at ice-sheet modellers, it is a little bit thin on results. It would be beneficial for the manuscript if the authors write it with a specific audience in mind. Regardless of that, the text has to be much more clear that the described results are results of simulations, and are not expected contributions of Antarctica to sea level. This seems like an obvious, and redundant comment, however, considering a high profile of this manuscript (a most like reference for the next IPCC report), its language and wording has to be precise. I would recommend to modify all statements similar to "The contribution of the Antarctic ice sheet..." (p.2 line 6), "East Antarctica mass change... " (p.2 line 9), etc. to "The projected contribution of the Antarctic ice sheet...", "Simulated East Antarctica mass change..." (I'm not going to mark all such phrases, but please correct them all).

We thank the reviewer for this review and all the suggestions. The climate community at large and the ice sheet modeling community are both the target audience of this article. We worked on the paper to clarify it, explain any unstated assumption in the previous version and highlight that these results are modeled or simulation results that widely depend on the assumptions made for the experiments throughout the text. We therefore think that a wide audience will be able to understand the results presented here. Providing more details on the results is beyond the scope of this manuscript: there is already a significant amount of results presented in the current version, and more detailed analysis of specific processes will be the subject of future ISMIP6 studies that are starting to be planned, so there has been only limited new analysis added in the text.

Another general comment, which is easy to address, is that it'd be better to use CMs (climate models) instead of AOGCMs. "CM" in CMIP5 stands for "Climate Model", additionally later in the text (lines 85-95) "CM" and "ESM" in names of the models, which outputs were used, indicate the type (complexity) of the model - either a Climate Model or Earth System Model.

We thank the reviewer for this suggestion. In order to be consistent with the other ISMIP6 publications [e.g., *Nowicki et al.*, 2020; *Barthel et al.*, 2020; *Jourdain et al.*, under review] that all use AOGCM, we decided to keep the AOGCM terminology. However, we reduced the number of times this acronym

is used and removed the acronym wherever it was possible. We also explained that the forcing comes from both Climate and Earth System Models.

Overall, the text has too much jargon (e.g. SMB; it's not clear what "idealized surface mass balance" means). The titles of sections and subsections are too cryptic (e.g. "ctrl_proj", "NorESM1-M RCP 8.5 scenario"). They need to be informative enough to give a general idea of section or subsection content. The subsection "2.1.4 Ice shelf collapse" is misleading in both its title and justification of the experiment. Perhaps it should have quotation marks to indicate the name of an experiment. Lines (157-159), state that hydrofracturing is the main mechanism that leads to an ice-shelf collapse. Though collapse of the Larsen B ice shelf preceded by surface melting, and hydrofracturing was specifically proposed to explain its collapse, it is not the only ice shelf to collapse, and collapses of other ice shelves, for instance Willkins Ice Shelf, were most likely unrelated to hydrofracturing or surface melting (it happened during austral winter). Because hydrofracturing is essentially the only mechanism that can be parameterized in an ice-shelf model, it does not mean that it is the only possible mechanism to trigger an ice-shelf disintegration. This part of the text needs clarifications.

The "idealized surface mass balance" was referring to the initMIP experiments and is now explained in the text. We changed the titles of several sections to be more informative, consistent and to reflect the content of the sections, and reorganized sections 2 and 3. Regarding the ice shelf collapse, a lot of research is still ongoing to better understand why and how ice shelves do collapse. The hydrofracturing is one of the mechanisms proposed and certainly does not explain all the collapses observed so far. However, this is one of the mechanisms that has been studied and is used in ice sheet models. It is therefore important to assess its potential on a large variety of ice flow models, as is done in the present manuscript. We modifiid the text to emphasize that hydrofracturing is only one possible mechanism that can explain ice shelf collapse.

As mentioned above, the manuscript documents experimental protocols and describes projected Antarctic contributions to sea level. It is unclear whether there will be in-depth analyses of this MIP. Considering the diversity of participated models, it would be interesting to know whether some useful lessons (apart from projected sea-level magnitudes and their spreads) could be learned from this exercise. For instance, because the initial ice sheet geometry affects ice flow and and ice discharge through the grounding line at the later times, it would be interesting to know whether ice-sheet models that use long spin-up as initialization, simulate significantly different ice discharge compared to ice-sheet models used present-day configurations as initial conditions. The same question applies for parameterizations of basal sliding - do models that use inversions of the present-day observations produce different results compared to models that don't employ inverse methods? These are rather suggestions, and it up to the authors to decide what is the scope of the manuscript.

The present manuscript is the first manuscript analyzing the ISMIP6 Antarctic results and its main focus is to investigate the potential sea level contribution from the Antarctic ice sheet over the 21st century, which is already a considerable amount of information. All the results will be made publicly available along other CMIP6 results to the public, and we expect that additional analysis will be performed in order to investigate in more details the role of varying processes, such as those suggested here. Some responses to the question of the impact of initialization method procedure can be found in the initMIP manuscript [*Seroussi et al.*, 2019].

Similarly, the structure of the manuscript is the authors' decision. The current version has a fairly lengthy description of sub-ice-shelf melting simulation. It is not entirely clear why this process has such a prominence compared to other, no better constrained processes (e.g. basal sliding, calving, etc.). As a suggestion, the authors might consider moving details to an appendix, and the current appendix (C at least) to supplemental online materials. The manuscript will benefit from streamlining. Currently, section 2 combines together various unrelated aspects (i.e. kinds of forcing, experiments, etc.). Having a better structure will improve the manuscript readability.

We thank the reviewer for these suggestions to improve the readability of the manuscript. We reorganized sections 2 and 3, and improved the sections' names. Unknows in basal melt rates and ocean conditions are a large problem to force ice sheet models and a large source of uncertainties, which explains the length of the discussion of this process.

Figures could be more illustrative. Overall, bar figures are difficult to read, simply displaying them in the models' alphabetical order is not informative. The authors might consider modifying the time axis in Fig. 1 (e.g. having uneven spacing prior to 1990s or so) to focus more on a period of time that have results from more models. Panels (b) and (c) in Fig. 3 seem to show the same field but in different units, the purpose of that isn't clear. Perhaps using log scale (for negative values it could a different colormap of the absolute values) in figs 6 and 14 might show better spatial variations on the grounded ice sheet, as the largest magnitudes are on ice shelves or in their immediate vicinity. It is unclear how to read fig. 15, its caption does not help with that.

The time axis on Fig.1 indeed leads to a lengthy pre-2015 period during which one 1 model produced simulations, so we changed the axis to better focus on the period during which more models provided results, starting in 1950. Figure 3 (b) and (c) show the same values but panel (c) uses a log scale to highlight the large uncertainties in fast flowing areas. We removed it as both reviewers find it confusing. Figure 15 has been improved to better show the basin contributions and the caption has been changed.

**2 Reviewer #2 (Anomymous)**

I commend all authors and contributors for their efforts and time investment into this MIP (one of many) and highly recommend this community effort for publication in TC. I have no significant points of concern; my only main comment is about the discussion and conclusions. Despite the tricky task of analyzing outputs from such a diverse range of forcings and model designs, I would have liked to see a stronger emphasis on the 'lessons learnt' from this exercise, and suggestions for possible ways forward. In my opinion, one of the key messages that should prevail from this MIP is that, despite the large spread in projections, significant advances have been made in the recent decade to reduce the uncertainties. Although this is mentioned in the text, I think this success should be stressed more and perhaps even quantified (e.g. L501 and following, L538-539). Moreover, as this is very much 'work in progress' whilst the modelling community continues its efforts to improve models, these MIPs are a great way to guide such improvements. Individual groups will have used ISMIP6 and related MIPs to test and upgrade their models, and other developers/users might benefit from adopting these improvements in their own models. Perhaps there is scope for a paragraph or two in the discussion on i) recent key challenges (numerics, physics,...) that have been considered/overcome by individual contributors and how this has influenced their results, ii) an expert judgement on key improvements that need to be prioritized in the near future? In light of future publications, such as additional results based on CMIP6, the community might also want to think about more concrete 'measures of progress'.

We thank the reviewer for this careful review and constructive comments. We better highlighted the progresses made since previous comparable efforts. Providing guidance to the community on the key challenges and improvements needed is indeed something very important and we detailed the paragraph in the discussion discussing these limitations to highlight the lessons learned from this MIP.

Below is a list of more specific comments and points for further clarification. L1 It might be worth introducing an abbreviation for Antarctic Ice Sheet, as I counted 10 instances on the first 3 pages alone.

Papers that have too many acronyms tend to be less readable, so we are limiting the number of acronyms used. However, we agree there are many instances of the "Antarctic Ice Sheet", many of which are not necessary, so we removed them when unnecessary.

L3 'estimated' → estimates of?

Done

L3 You say 'primarily because of differences in the representation of physical processes and the forcings employed' but my understanding is that the initial

state of the ice sheet and numerical design of the models are equally important sources of uncertainty?

Indeed, all these factors are important sources of uncertainty and their relative importance remains to quantify. We changed the text.

L4 13 groups?

Done

L7 '...between -7.8 and 30.0cm...': is this for a fixed forcing scenario, or does it include uncertainties from variability in models and the full range of forcings?

Done

L15 define AOGCM

Done

L15 'overall' → additional

Done

L27 'paradigm shift' is rather vague. Perhaps you can be more precise, e.g. by saying that models have been verified against analytical solutions of ice flow, grounding line flow etc.

Done

L30 Do you mean that model validation against observations of past changes is critical to improve projections, or are you alluding to a more general understanding of how climate change affects sea level?

We need to understand the processes that caused the recent past changes of the ice sheets and to be able to reproduce them if we want to improve our confidence in future ice sheet projections. We clarified that.

L33-35 Perhaps the ice sheet initial state (and results from initMIP) should be included here as an additional source of uncertainty.

Very good point, we added the initial state as another source of uncertainty.

L41 'mitigate the gaps' seems like an unfortunate choice of words. My understanding is that MIPs aim to quantify the spread in model projections, rather than to eliminate the spread?

Rephrased

L44-48 I was expecting to read about the impact of the initial ice sheet state here, but instead the focus is on SMB. Can you provide a 1-line summary of the initMIP results?

Done

L135 Perhaps you can point out that this result was obtained in the context of the idealized MISOMIP experiment, but has not been tested for realistic geometries.

We added that point.

L140-145 Although Jourdain et al. (under review) will provide further details, it would be nice to have a little bit more information here. E.g. it is not clear what is meant by 'random samplings of Antarctic melt rate and ocean temperature'. Are these melt rates from Rignot et al., and is the ocean temperature taken from observations/reanalysis?

We added details for the source of these different observations.

L185-186 I'm unclear about the difference between ctrl and ctrl_proj. Are they identical except for the duration, i.e. ctrl runs from the initialization time until 2100, whereas ctrl_proj runs from 2015 until 2100?

The ctrl experiment is similar to the initMIP-Antarctica results and starts from the model's initial state, while the ctrl_proj experiments start in 2015. We added details about the difference between these two experiments.

Table 1, first row. Is 'Ocean coefficient' the $\gamma_0$ parameter in Eq(1) and what is meant by 'Low', 'Medium' and 'High'?

Yes, this refers to the values used for the $\gamma_0$ parameter in Eq(1) and represents the Median, 5% and 95% values of the distribution. We changed the 'Low', 'Medium' and 'High' values to 'Median, '5%' and '95%' to provide more accurate information (as is used in Fig.12).

L201 It would be good to have some further info about the 'open experiments' here. Does it mean that some ocean conditions (T,S,. . .) are prescribed but the melt parameterization is left free?

Yes, all parameterizations have to use the same ocean conditions provided from the CMIP models, but they melt parameterization used differ between the models and are listed in Table 3. We added a reference to Table 3.

L217 include abbreviations FE and FV to help the reader interpret the second column of Table 3.

Done

 what is meant by 'consistent' here? Perhaps this can be quantified.

Done. We also added scales on Fig.2 to facilitate comparison

L278 'ice shelves that extend slightly farther': again, perhaps this can be quantified. Could this be a resolution issue, i.e. the offset is on the order of the resolution of the analysis mesh?

The second part of the sentence provides more quantitative information. We also added scales on Fig.2 to facilitate comparison.

L283-285 this information seems to be repeated from lines 268 and 270.

Indeed, we removed the information on lines 268-270.

L300 'trends cannot be considered as a physical response of the AIS...': despite the constant climate conditions applied, could internal ice dynamics not give rise to a trend?

Part of the trend in indeed caused be ice dynamics and response to climate forcing applied, but part of it is also caused by the response to initial conditions, which is why this trend should not be considered strictly as a physical response of the ice sheet to climatic conditions.

L309 The reference to figure 1 is appropriate here, but I'm finding it hard to distinguish the individual model results due to the choice of colour scheme. It is therefore difficult to verify this point.

We added data for the evolution of ice volume and ice volume above floatation (as well as evolution of ice extent and ice shelves extent) in table B2 to facilitate comparison between models.

L310 please check this sentence as I'm not sure what is meant here.

Done

L312 At this point it is unclear why NorESM1-M was singled out for these experiments. Can you comment?

We wanted to analyze the response of ice flow models for one specific experiment, to show the variety of response, understand why and how the results differ, which is why we analyzed one experiment in more details.

L325 'slit'?

split. Done

 The Siple Coast ice streams seem to produce an equally large response, yet they are not mentioned here?

Most of these changes are caused by the models that have grounding lines extending further than the present-day grounding line in this region, so we don't think this should be considered as ice stream changes, but rather ice shelf changes.

 4 out of 6?

Done

 You say that 'uncertainties for the WAIS are larger than for the EAIS' but I can't see any significant difference between the length of the error bars in Figure 8. . .

The uncertainty is on the same order of magnitude, and models with more changes in ocean conditions have larger uncertainties.

 Both here and later on, it would be useful to reference back to Table 3 with the experiment names, e.g. '...experiments were simulated with both open (exp01-04) and standard (exp05-08)...'

Done

 Again, a reference to the experiment names in Table 3 would be helpful, i.e. exp05.

Done

 superscript st in 21st

Done

 Are ocean processes even reliably included in the Greenland studies?

The impact of the ocean is calibrated from past observations of ocean conditions and ice front retreat rates, and extended into the future based on climate models' outputs. However, climate models do not simulate ocean conditions in fjords, which is in a way similar to the problems we have in Antarctica with ice shelf cavities not being included, but was treated differently as ice front positions are forced in the Greenland simulations.

L499-500 As discussed earlier, it would be nice to see a more quantitative statement here, and further documentation on what is meant by 'significant improvement'.

We detailed this paragraph

L538 In my opinion this sentence is somewhat misleading. I assume that the 'main sources of uncertainties' refer to the uncertainties that were addressed as part of this study? Other sources of uncertainty such as the initial state of the ice shelf were not addressed here, and could be equally important.

Rephrased the sentence to avoid this ambiguity.

Table A1. title: FL instead of FX?

Done

Table B1. title: add (2015) to better specify 'beginning of the experiments'?

Done

Figure 2. Something gone wrong with the colorbar in panel a? Also, black lines are very hard to see with the dark blue background, so consider adjusting the colors for more contrast.

Fixed problem with colorbar in panel a. We kept the black lines to be consistent with the initMIP figure and because white or other shades of grey does not improve the contrast.

Figure 3. Yellow text is hard to read. I'm not sure what the log plot in panel c contributes to the analysis. Spatial maps of ice thickness and velocity std between models might be a useful metric to identify areas where models disagree the most and highlight geographical regions where efforts for improvement should be focused.

Yellow was changed to a darker shade be easier to read. We agree that the Log plot of the velocity does not add much information and was removed. We did not add the spatial plots of thickness and velocity std as this is beyond the scope of this manuscript that mostly aims at looking at projections of future Antarctic evolution.

Figure 12 and 13. Experiment names in the legend would be a handy cross-reference to Table 3 here.

Done

 Why is the sea level contribution larger (more negative) without ice shelf collapse?

It is the opposite, there is actually more mass gain in the absence of ice shelf collapse than with ice shelf collapse. The sign is negative, so it means mass gain, and there is less mass gain in the presence of ice shelf collapse. We added a note in the legend to avoid confusion.

Figure 15. It is very hard to distinguish individual basins here, whereas this is crucial to understand the figure. Perhaps consider splitting into subfigures with equal axis to show results for different basins or groups of basins? Also consider adding basin names to help readers understand the main text (L473-475).

This is a great suggestion, and we separated the figure into WAIS, EAIS and Peninsula.

[revised manuscript text omitted]